# Learning from Yeast about Mitochondrial Carriers

**DOI:** 10.3390/microorganisms9102044

**Published:** 2021-09-28

**Authors:** Marek Mentel, Petra Chovančíková, Igor Zeman, Peter Polčic

**Affiliations:** Department of Biochemistry, Faculty of Natural Sciences, Comenius University in Bratislava, Mlynská dolina CH-1, Ilkovičova 6, 842 15 Bratislava, Slovakia; marek.mentel@uniba.sk (M.M.); petra.chovancikova@uniba.sk (P.C.); igor.zeman@uniba.sk (I.Z.)

**Keywords:** budding yeast, *Saccharomyces cerevisiae*, mitochondria, transport, mitochondrial carrier family, human disease

## Abstract

Mitochondria are organelles that play an important role in both energetic and synthetic metabolism of eukaryotic cells. The flow of metabolites between the cytosol and mitochondrial matrix is controlled by a set of highly selective carrier proteins localised in the inner mitochondrial membrane. As defects in the transport of these molecules may affect cell metabolism, mutations in genes encoding for mitochondrial carriers are involved in numerous human diseases. Yeast *Saccharomyces cerevisiae* is a traditional model organism with unprecedented impact on our understanding of many fundamental processes in eukaryotic cells. As such, the yeast is also exceptionally well suited for investigation of mitochondrial carriers. This article reviews the advantages of using yeast to study mitochondrial carriers with the focus on addressing the involvement of these carriers in human diseases.

## 1. Introduction

Mitochondria of eukaryotic cells are surrounded by two biological membranes. The outer membrane that separates mitochondria from the cytosol is permeable for solutes due to the presence of voltage-dependent anion channels (VDAC). The inner membrane, conversely, is required to be impermeable for protons and ions as it is the site of electrochemical (chemiosmotic) coupling between complexes of the respiratory chain and ATP synthase. The solutes that are required to penetrate the mitochondrial matrix or be exported to either the cytosol or mitochondrial intermembrane space are transported across the inner membrane by a system of highly selective transporters, many of them structurally homologous, belonging to the mitochondrial carrier family (MCF). As transport of these molecules is necessary for numerous metabolic processes that occur both in the mitochondria and the cytosol, the proper function of transporters of the inner mitochondrial membrane is of vital importance for cell physiology and health. It is thus not surprising that in humans, there is a considerable number of diseases resulting from defects in mitochondrial transport.

Yeast *Saccharomyces cerevisiae*, also referred to as budding yeast or baker’s yeast, have been in service to mankind since ancient times. Many traditional and modern food producing technologies owe to their ability to ferment sugar. Being used as a model of choice for scientific research since the very beginning of modern science, yeast has become a traditional eukaryotic model organism with an unprecedented impact on biochemistry and cell biology [1]. It has been employed to characterize countless fundamental processes in eukaryotic cells, including gene expression, the cell cycle progression, signal transduction and membrane trafficking, to name just a few. One of the areas in which yeast has proven to be an especially fruitful model is the area of mitochondrial research and bioenergetics.

There are numerous aspects of yeast biology that make yeast a model of choice for studying mitochondria. Budding yeast is a single cell eukaryote that is easy to cultivate in chemically well-defined media. Due to its short doubling time, with approximately 90 min for a wild type in rich media and low requirements, yeast can be grown in the laboratory to significant quantities. With the protocols that are available for isolating mitochondria from yeast cells, one is thus able to prepare isolated organelles in a required amount, purity, and quality (e.g., mitochondria with well coupled oxidative phosphorylation due to the intact membranes) for biochemical experimentation.

Due to the extremely effective homologous recombination, *S. cerevisiae* are prone to genetic manipulation. Precisely aimed specific gene disruption and introduction of mutant versions of genes are easy tasks in yeast, unlike in many other traditional eukaryotic models. In 1996, budding yeast became the first eukaryotic organism of which the nuclear genome was completely sequenced [2]. Various genome scale analyses have been since possible, including a genome-wide deletion project [3] and synthetic lethality screens [4]. Collections of deletion mutants with single deletions of any nonessential open reading frames are either available for experimentation [3], or can be easily prepared in any genetic background [5,6].

## 2. Mitochondrial Metabolism and Genetics

One of the reasons *S. cerevisiae* has become a model organism of choice is its robustness when it comes to (energy) metabolism, consequently enabling the study of associated cytological changes at molecular and genetic levels. This robustness is often connected to the characteristics of its mitochondria. *S. cerevisiae* is able to tolerate extensive deletions of wild-type (ρ^+^) mitochondrial DNA (mtDNA), leading to the formation of ρ mutants [7,8] If there is no mtDNA present, these mutants are referred to as ρ^0^ to distinguish them from ρ^–^ mutants harbouring remnants of mtDNA in their mitochondria. Contrary to wild-type strains, ρ mutants are unable to grow on respiratory (non-fermentable) carbon sources, e.g., ethanol, lactate or glycerol, and when grown on such medium supplemented with limiting amounts of fermentable carbon source (e.g., glucose, galactose, maltose, sucrose), they form colonies of reduced size. This growth phenotype led to the widely used designation of these mutant strains—*petite* strains [9].

There are still other classes of respiratory-deficient mutant strains of *S. cerevisiae*; *mit*^−^ mutants carry point mutations in mtDNA while *pet* mutants are characterized by the mutations in nuclear genes involved in mitochondrial functions [9]. Fungal mitochondria are involved in a plethora of biological processes including oxidative phosphorylation, metabolism of lipids, carbohydrates, amino acids and nucleotides, biosynthesis of haem, Fe-S clusters and ubiquinone, protein translocation, RNA processing, translocation of nucleotides (e.g., ADP for ATP and vice versa) [10]. Considering all of these processes, only a handful of mitochondrial proteins have been shown to be essential for growth of yeast cells, reflecting the robustness of *S. cerevisiae* mentioned above. These proteins fall into two categories; most of the essential mitochondrial proteins are involved in the biogenesis of the organelle (e.g., in import of proteins into mitochondria), while a substantial number of the essential mitochondrial proteins is required for the biogenesis of Fe-S clusters [11]. This is consistent with the view that biogenesis of Fe-S proteins is the only essential function of mitochondria in eukaryotes [12].

Another interesting feature of *S. cerevisiae* is its ability to grow under anaerobic growth conditions, hence its facultatively anaerobic nature. Once again, and maybe surprisingly, this ability is also connected to the functionality of mitochondrial protein, the transporter, as will be detailed below (see Section 3). When grown in strictly anaerobic conditions, addition of ergosterol to the cultivation media is required, as the biosynthesis of this membrane lipid is oxygen dependent [13,14]. However, only a minute, nanomolar concentration of oxygen (7 nM) enables both the synthesis of ergosterol and growth of the yeast cells [15]. Importantly, essential requirement for an anaerobic growth is the fermentable nature of utilized carbon source. As in the absence of oxygen—the terminal acceptor of electrons—mitochondria cannot be involved in the synthesis of ATP by oxidative phosphorylation, other means must be involved to supply both the cytosol and mitochondria with ATP. Phosphorylation linked to the cytosolically localised glycolytic pathway fulfils this role.

Interestingly, even when *S. cerevisiae* is grown aerobically on a fermentable carbon source, despite the presence of oxygen, a substantial portion of substrate is not fully oxidised to carbon dioxide and water but is rather partially oxidised to yeast fermentation product—ethanol [16]. This ability to suppress the respiratory energy metabolism in the presence of oxygen in favour of aerobic alcoholic fermentation at high growth rates is known as the Crabtree effect [17]. Generally, only after the exhaustion of fermentable carbon source do yeast cells switch to respiratory metabolism by the transition referred to as a diauxic shift.

This flexibility of the energy metabolism of *S. cerevisiae* provides the cells with twofold advantage in the competition for limited resources in its natural ecological niche. As the genus name hints, *Saccharomyces cerevisiae* competes for sweet seasonal angiosperm (flowering plants) fruits rich in energy carbon. Starting the competition with aerobic fermentation, the Crabtree-positive yeast takes advantage of the fast generation of ATP by fermentation, resulting in greedy consumption of fermentable resources and ultimately allowing rapid reproduction of cells [18]. This is still only half of the explanation of the yeast’s evolutionary success. The remainder is represented by the yeast fermentation end-product—ethanol. While ethanol is toxic for most microorganisms, *S. cerevisiae* tolerate high concentrations (some strains up to 14%) [19]. Furthermore, to make the whole strategy perfect, after poisoning, the competitors and undergoing diauxic shift *S. cerevisiae* can utilize this fermentation product as a respiratory substrate for mitochondrial oxidative phosphorylation. This tactic was accurately described as the “make–accumulate–consume” strategy [19,20] or more metaphorically as “hold your breath, hog the loot, poison witnesses, burn the evidence” [21].

## 3. ADP/ATP Carrier

As mentioned above, easily available methods of yeast genetics make yeast a model, in which a genetic approach to bioenergetics can be employed. This was first recognized in the late sixties of the twentieth century, when the traditional models for mitochondrial research were animal tissues rich in mitochondria, mostly muscles, such as bovine heart. The first yeast mutant, later shown to have a mutation in gene encoding for mitochondrial carrier, was mutant with atypical oxidative phosphorylation, designated as *op1*. It was prepared by chemical mutagenesis of wild-type yeast [22] and grew extremely slowly on non-fermentable carbon sources, although it was able to maintain almost normal rate of respiration in the presence of uncoupler. Analyses of mitochondria isolated from the *op1* mutant cells, including the response of parameters of respiration to bongkrekic acid and atractyloside, specific inhibitors of mitochondrial ADP/ATP transport, indicated that the mutation may affect the ADP/ATP translocator of inner mitochondrial membrane [23,24], a protein whose existence in mitochondria was suggested earlier, when transport of ATP and ADP was initially investigated using rat liver mitochondria [25,26]. Genetic screening of the yeast genomic library later resulted in identification of the first gene encoding for mitochondrial carrier protein—*AAC1* [27], (at the same time when the sequence of bovine ADP/ATP carrier was determined by protein sequencing [28]) and the second gene encoding for a major yeast isoform of the carrier—*AAC2* [29]. The *op1* mutation was shown to be a single amino acid substitution in Aac2p (Arg96His) [30]. Finally, the third yeast isogene encoding for ADP/ATP carrier—*AAC3*—was cloned after the analysis of *op1* revertant able to grow on glycerol, which resulted from the recombination between defective *aac2**^Arg96His^* and *AAC3* [30].

The principal function of ADP/ATP carrier is the equimolar exchange of ATP and ADP across the inner mitochondrial membrane (Figure 1). This function is required for the survival of yeast when grown either under anaerobic conditions or on non-fermentable carbon source.

The requirement of the functional ADP/ATP transport for anaerobic growth stems from the essentiality of the intramitochondrial ATP for the survival and growth of eukaryotic cells [31,32,33]. When cells are grown in the absence of oxygen, intramitochondrial ATP cannot be maintained by oxidative phosphorylation. Instead, the ATP generated by the substrate-level phosphorylation during the fermentation in cytosol needs to be imported to mitochondria. Import of cytosolic ATP to mitochondria, coupled to export of mitochondrial ADP into cytosol is, indeed, catalysed by ADP/ATP carriers [34].

In yeast, the same direction of the transport, import of ATP into mitochondria and export of its hydrolytic product ADP, can also be observed under aerobic conditions in the presence of fermentable carbon source (Crabtree effect, see chapter 2) [35]. The direction of ATP transport thus differs from that following the diauxic shift from fermentative to respiratory metabolism [16,17], when ATP synthesized in mitochondria by oxidative phosphorylation is exported from mitochondria to supply the cytosol in exchange for ADP that is imported to mitochondria [36]. Naturally, the transport of ATP by ADP/ATP carrier from mitochondria is required to support the growth on ethanol, lactate or any other respiratory (non-fermentable) carbon source *S. cerevisiae* is able to utilize, as catabolism of these substrates does not result in the net ATP synthesis in the cytosol.

The three *AAC* isogenes in *S. cerevisiae* encode for functionally identical proteins, which are, depending on growth condition, distinctly regulated at the level of transcription. While *AAC1* is expressed very weakly and does not actually contribute to ADP/ATP transport under any known conditions, *AAC2* represents the major isoform expressed in aerobically grown cells and *AAC3* is expressed in anaerobiosis [37,38,39]. This is reflected by different phenotypes of individual *AAC*-deletion strains. There is no growth phenotype associated with deletion of *AAC1*. *Δaac2* strains fail to grow on non-fermentable carbon sources, as in the absence of carrier, intramitochondrial ATP generated by oxidative phosphorylation cannot be transported to the cytosol and the rest of the cell. Since, as mentioned above, the growth under anaerobiosis requires the functional ADP/ATP transport across the mitochondrial membrane to supply the mitochondrial matrix with the ATP produced in fermentation in the cytosol, the absence of functional ADP/ATP carrier results in growth failure in the absence of oxygen. Because expression of *AAC2* ceases only at very low oxygen concentrations, deletion of *AAC3* contributes to a detectable phenotype—absence of growth—only in strict anaerobiosis or in double *Δaac2Δaac3* mutant under conditions of low oxygen.

While the first hints on membrane topology of mitochondrial carriers came from analyses of proteins isolated from mammalian mitochondria [40,41,42,43,44], functional analyses of yeast AACs and ultimately the crystallisation of bovine protein led to the detailed model of its membrane topology, 3D structure and generalized model carrier of MCF (see [45,46] for review and Figure 2).

### 3.1. Human ADP/ATP Carrier

In humans, four isoforms of ADP/ATP carrier, referred to as ANT (adenine nucleotide translocator, ANT1, ANT2, ANT3 and ANT4), have been characterized [50,51], and ANT has been implicated in several mitochondrial disorders [52]. It is thus of great interest to study the biochemical properties of corresponding mutant proteins. Yeast cells have been efficiently used to study human ADP/ATP carriers for two decades now.

ANT1 (SLC25A4) encodes for the main human isoform present in heart, skeletal muscle and brain [51,53]. Heterologous expression of the native human ANT1 in yeast resulted in a low amount of protein in mitochondrial membrane that could not support the growth of *Δaac1Δaac2* double mutant strain on non-fermentable carbon source (glycerol). Its expression was, however, considerably improved and the growth of the yeast strain was restored when eleven N-terminal amino acids of human protein were replaced with the 26 residues of the corresponding region of yeast Aac2p. The transport rate in the yeast mitochondria containing such chimeric ANT1 (yNhANT1) was shown to be similar to that of bovine heart mitochondria with native bovine AAC1 and, sensitive to carboxyatractyloside and bongkrekic acid [54].

A similar technique was employed to express other human isogenes in yeast. While expression of chimeras derived from three isoforms—yNhANT1, yNhANT2 or yNhANT3—complemented the defect in *Δaac1Δaac2Δaac3* strain, supported its growth on glycerol and ethanol containing media (YPEG) and retained their ADP/ATP transport capacity, the chimera derived from the last discovered human ADP/ATP carrier, a testis-specific isoform also present in the liver and brain, ANT4 (SLC25A31)—yNhANT4—did not [50]. Moreover, the overproduction of yNhANT4 from multi-copy plasmid interfered with cell growth on fermentable carbon source, which was not observed with yNhANT2. Ethyl methanesulfonate (EMS) mutagenesis led to the identification of three independent yNhANT4 mutations (Ala30Val, Pro95Ser, Ser202Leu) that resulted in the increased amount of carrier in the mitochondria and restoration of respiratory growth. Based on a comparison with the crystal structure of bovine ANT1 [49], it was suggested that these mutations in the transmembrane domains may affect the interaction of the yNhANT4 with lipids. When ADP/ATP exchange was measured in the mitochondria isolated from yeast expressing human yNhANT1, yNhANT2, yNhANT3, three mutant yNhANT4 and yeast Aac2 carriers, the similar K_M_ for ADP in micromolar range was observed for all of them [55].

To identify individual ANT specific inhibitors with therapeutic potential, a high-throughput screening of 65,000 compounds using yeast strains [55] expressing individual human ANTs was performed. Cell-permeable ANT inhibitors, which reduced the growth of yeast cells expressing one of the four human ANTs, were identified and ADP/ATP exchange assays in isolated yeast mitochondria with selected drugs were performed [56]. Closantel is halogenated salicylanilide with antiparasitic activity [57] and CD437 is synthetic retinoid that selectively induces apoptosis in cancer cells [58]. They were identified by this methodology as broad-spectrum ANT inhibitors, whereas natural diterpene amine known as anticancer compound leelamine [59] was found to be a modulator of ANT function. Authors suggest that, for example, selective inhibitors of ANT4, which are exclusively expressed in testis, may be optimized for use as male contraceptive agents [55].

On the other hand, the number of commonly used drugs may affect the mitochondrial carriers as off-targets. Complete human ANT1 or its truncated version Δ1-10ANT1 without the first 10 residues were expressed in the *Δaac1Δaac2* yeast strain to study drugs inducing inhibition of human ADP/ATP carrier [60]. Of two tested versions, the Δ1-10ANT1 rescued growth of the deletant strain. Overexpressed human ANT was purified and used in thermostability shift assays and transport measurements in proteoliposomes to investigate effect of inhibitors. Among the tested drugs, chebulinic acid, CD437 and suramin were the most potent inhibitors of ANT1 with IC_50_-values in micromolar range. Chebulinic acid is ellagitannin, a plant polyphenol with anticancer activity [61]. Suramin is a bis-polysulfonated naphthylurea used for treatment of trypanosomiasis [62] as well as an antineoplastic agent used as a chemosensitizer [63,64].

### 3.2. Autosomal Dominant Progressive External Ophthalmoplegia (adPEO)

Autosomal dominant progressive external ophthalmoplegia (adPEO) is a rare human adult-onset disease with multiple deletions of mtDNA (OMIM 609283). It is characterized by ptosis, ophthalmoparesis, exercise intolerance and mild reduction of activity of respiratory complexes I, III and IV [65]. ANT1 mutation Val289Met was identified in one sporadic patient and Ala114Pro mutation in four families and in one sporadic patient with adPEO [66,67]. Functional analysis of ANT1 mutations in human cells lines is problematic as ANT1 is not expressed in commonly used cultured cell lines [66] and overexpression of wild-type ANT1 induced apoptosis is in human cells [68]. Functional consequences of Ala114Pro mutation on function of ANT1 were therefore investigated in *Δaac1Δaac2* and *op1* (*aac2^Arg96His^*) yeast strains, in which *AAC2* containing a corresponding mutation—Ala128Pro—was expressed. Ala128Pro mutation was shown to cause a significantly reduced growth on glycerol medium, which results from defective ADP/ATP transport, as neither large-scale rearrangements nor depletion of mtDNA was observed [66].

In another study, it was shown that the same mutation results in depolarisation, structural swelling, and disintegration of mitochondria, and ultimately in an arrest of cell growth. The abolished import of proteins into mitochondria was documented by following the import of non-proteolytic ATPase of the AAA family—Mcx1p—fused with GFP. It was suggested that Ala128Pro mutation in Aac2p induces an opening of an unregulated channel that allows a free passage of protons and other solutes across the inner membrane [69].

Human ANT1 mutants Ala114Pro and Val289Met were also directly expressed and studied in yeast. Mutant carriers were unable to restore growth of the yeast *Δaac1Δaac2Δaac3* strain on lactate medium. The corresponding genes were transcribed, but proteins were not detected in yeast mitochondria neither with specific antibodies nor by [^3^H]-atractyloside binding assay [70]. Authors thus suggested that similar a situation may occur in human cells affected with adPEO.

Other mutations in human ANT1 were identified in patients with adPEO: missense heterozygous Leu98Pro in a Greek and Italian families [71,72], Asp104Gly in Japanese family [73] and Ala90Asp in a German family [74].

ANT1 Leu98, Ala114 and Val289 have corresponding amino acids in Aac2p Met114, Ala128 and Ser303. These ANT1 mutations were indirectly studied in yeast *Δaac1Δaac2* strain expressing Aac2 variants Met114Leu, Met114Pro, Ala128Pro, Ser303Val and Ser303Met. While the “humanized wild-type” Aac2p Met114Leu and Ser300Val supported the growth of the cells on non-fermentable carbon sources, the growth was markedly impaired with Aac2p Ala128Pro and Ser303Met and the defect was most severe with Met114Pro [75]. Aac2p mutations equivalent to adPEO-associated mutations caused reduction of the content of mitochondrial cytochromes, cytochrome *c* oxidase activity, cellular respiration and the defect of ADP/ATP transport.

ANT1 Asp104Gly and Ala114Pro, adPEO-associated mutations, were studied in yeast when chimeric human ANT1 (containing yeast N-terminal sequence) yNhANT1 [54] with corresponding substitutions were expressed in *Δaac1Δaac2* strain. Cells expressing yNhANT1-Asp104Gly displayed strongly reduced growth on media with ethanol as compared to cells with yNhANT1 without mutation. Transformants with yNhANT1-Ala114Pro did not grow on the same media. The viability of *petite*-negative *Δaac1Δaac2* strain with ANT1-Asp104Gly was significantly reduced, probably due to instability of mtDNA, a phenomenon observed also in human cells with ANT1 Asp104Gly mutation [76].

Yeast Aac2p adPEO-type mutations Ala128Pro, Ala106Asp and Met114Pro correspond to human ANT1 Ala114Pro, Ala90Asp and Leu98Pro mutations. All these Aac2p adPEO-type mutations share common dominant phenotypes, such as damage of respiratory chain, synthetic lethality with low transmembrane potential (Δψ) conditions induced by absence of mitochondrial protease Yme1p, hypersensitivity to uncoupler CCCP and mtDNA instability. Interestingly, the Aac2 Ala137Asp mimicking ANT1 Ala123Asp mutation linked to mitochondrial myopathy and cardiomyopathy [77] exhibits similar dominant phenotypes leading to the loss of cell viability. These four pathogenic mutations occur in a dynamic gating region on the cytosolic side of the protein and all of them cause the uncoupling of mitochondrial respiration, probably by increasing the inherent proton-conducting activity of the carrier [78].

Aac2p Ala128Pro mutation was also shown to dominantly cause the ageing-dependent mitochondrial degeneration, decrease of Δψ and shortened replicative lifespan. Growth of the cells is inhibited at 25 °C in a dominant–negative manner independently of nucleotide exchange activity of the mutant protein. The reduced viability and mitochondrial degeneration of Aac2-Ala128Pro-expressing cells can be suppressed by caloric restriction and by longevity mutations, including *Δgpr1*, *Δtor1*, *Δsch9*, *Δrei1*, *Δrpl6B*, *Δtma19*, *Δrpd3* and *Δrpl31A*, which are all known to reduce cytosolic protein synthesis [79].

### 3.3. Autosomal Recessive mtDNA Depletion Syndrome 12b

A recessive mutation in ANT1, Ala123Asp was linked to childhood onset mitochondrial myopathy and hypertrophic cardiomyopathy with exercise intolerance and lactic acidosis (OMIM 615418) with no ophthalmoplegia. Skeletal muscle of patient showed ragged-red fibres and multiple deletions in mitochondrial DNA in muscle [77]. ADP/ATP transport was completely absent in proteoliposomes reconstituted with protein extract from the patient’s muscle. Aac2p Ala137Asp mutation mimicking ANT1 Ala123Asp was investigated after expression from centromeric vector in the yeast *Δaac1Δaac2* strain. The cells expressing mutant versions of *AAC2* were not able to grow on non-fermentable carbon sources. Cytochrome spectra of glucose-grown cells displayed significant reduction of the cytochromes *b* and *aa_3_*. Proteoliposomes were reconstituted with extracts from cells with Aac2p Ala137Asp and ATP transport activity was not detected in contrast to control with wild-type Aac2p, resembling observation with patient muscle protein extracts. Mitochondria from mammalian ANT1-deficient cells have elevated amounts of ROS and accumulate mtDNA rearrangements [80]. When heteroallelic strain *aac1*, *aac2*/*aac2^Ala137Asp^* was exposed to ROS scavengers—dihydrolipoic acid or N-acetyl cysteine—its viability was noticeably increased, suggesting that anti-ROS therapy may be beneficial to patients [77].

The same mutation (Aac2p Ala137Asp), as well as three adPEO-type mutations (Aac2p Ala106Asp, Met114Pro and Ala128Pro) were shown to cause Aac2p protein misfolding. This perturbance in the inner mitochondrial membrane further negatively affects the assembly and stability of multiple protein complexes in the membrane, such as respiratory chain supercomplexes, inner membrane preprotein translocase complexes TIM22 and TIM23, and dimeric structure of ATP synthase, which ultimately inhibits cell growth. In contrast to Ala137Asp, in adPEO-type mutants, Aac2p aggregates were detected, indicating that different conformations are adapted by these mutants [81].

When the Aac2p Ala137Asp mutant lacking transport ability was expressed, in mitochondria it still interacted with yeast respiratory supercomplexes III_2_IV_2_ and III_2_IV. The complex IV–related defects in activity, and in the amount of mtDNA-encoded subunits Cox1, Cox2 and Cox3 are greater in mitochondria expressing the transport-inactive Aac2p Ala137Asp as compared with the complete absence of Aac2p. Absence of Aac2p function results in the dysregulation of mitochondrial translation of the subunits of complex IV and V. Functional Aac2p is thus important for normal mitochondrial translation and investigation of this regulatory mechanism may provide novel insight into human diseases associated with ADP/ATP carrier [82].

In a patient with mild mitochondrial myopathy, the recessive mutation was identified as Arg236Pro in ANT1 and from the predicted structure it was hypothesized that it could affect the folding of carrier [83]. Information about importance of Arg236 for function of the carrier came from the previous experiments in yeast, where substitution of equivalent Arg253 in Aac2p to isoleucine led to the disruption of transport function [84,85].

Homozygous mutation Gln218Pro in ANT1 was found in patient with myopathy, subsarcolemmal mitochondrial aggregations, cardiomyopathy, lactic acidosis, and l-2-hydroxyglutaric aciduria [86]. In this case, the corresponding residue in yeast Aac2p Trp235 is not conserved and studies of ANT1 Gln218Pro using a yeast model have not been published yet.

### 3.4. Autosomal Dominant Mitochondrial DNA Depletion Syndrome 12a

The third clinical phenotype with early-onset mitochondrial disease (OMIM 617184) characterized by severe combined mitochondrial respiratory chain deficiencies and marked loss of mitochondrial DNA copy number in skeletal muscle was caused by dominant mutations in ANT1: Arg80His and Arg235Gly [87]. It is important to note here that the mutation Arg80His in ANT1 is equivalent to the *op1* mutation (Arg96His) in Aac2p (see above).

Arg235, together with adjacent Arg236, the mutation of which was described above, represents the part of highly conserved motif RRRMMM, characteristic for ADP/ATP carriers. Substitutions of individual arginines in this motif in yeast Aac2p have been shown to abolish the transport activity of the carrier [84].

To validate the pathogenic role of ANT1 mutations in vivo, *AAC2* with corresponding mutations, Arg96His and Arg252Gly were expressed in the *Δaac1Δaac2* yeast strain. Alternatively, to evaluate the dominance/recessivity of mutations, the mutant versions of *AAC2* were expressed in the wild-type strain (W303-1B), which expresses also a genomic wild-type *AAC2* gene. Neither of the two mutants restored the growth of the *Δaac1Δaac2* strain on respiratory substrates. Aac2p Arg96His mutant was present in mitochondria in similar quantity as the wild-type Aac2p, but the detected levels of Arg252Gly were low. Neither of the two mutants displayed the dominant effect as both W303-1B transformants grew on glycerol comparably with the control. Both mutations caused decreased levels of cytochromes and markedly diminished respiratory capacity. Transport assays in liposomes fused with mitochondrial membranes showed a 40% decrease of ADP uptake for Arg96His mutant while no transport activity was detected for Arg252Gly mutant, likely due to the low expression. Arg96His-related phenotypes may be linked to impaired function of the mutant Aac2p because Arg80His alteration affects contact point I of the substrate-binding site. The Arg252Gly mutation abolishes one of the three salt bridge interactions of the matrix network, which may have additional functions in the biogenesis of the carrier [87,88].

Further investigations of Aac2p Arg96His and Arg252Gly mutations were performed with heteroallelic strains *AAC2/aac2^R96H^* and *AAC2/aac2^R252G^*, and both strains showed reduced growth on synthetic lactate medium compared not only to the homoallelic (*AAC2/AAC2*), but also to the hemiallelic strain (*AAC2/empty vector*) [89]. Both heteroallelic mutant strains have decreased respiratory activity and increased *petite* formation rate, reflecting the increased mtDNA instability, similarly as it is observed in the patients carrying ANT1 mutations Arg80His and Arg235Gly [87]. The observation that OXPHOS phenotypes in the heteroallelic strains were more affected than in the hemiallelic strain indicates that the dominance of the two mutations is not due to haploinsufficiency but rather due to a gain of function. This is further corroborated by the observation that the respiratory efficiency can be increased by addition of antioxidant agent N-acetylcysteine, but only in heteroallelic strains (*AAC2/aac2^R96H^* and *AAC2/aac2^R252G^*) [89].

Mild childhood-onset skeletal myopathy was linked with dominant mutation ANT1 Lys33Gln, which leads to OXPHOS defects in muscle [90]. Lys33 forms a salt bridge with the conserved Asp232 in the matrix network and Lys33Gln mutation would eliminate this interaction and also negatively affect the function of Gln37 that forms a highly conserved glutamine brace [90,91]. Lys33 is conserved among ANTs from metazoans, plants and fungi, and the corresponding residue in yeast Aac2p is Lys48. Neutralisation of this conserved positive charge by replacement of lysine with alanine abolishes the growth of yeast on glycerol, leading to a substantial decrease in mitochondrial ATP synthesis and resulting in almost a nondetectable electroneutral ADP/ADP exchange when the mutant protein is reconstituted in proteoliposomes [92,93].

### 3.5. Drug Screening

Yeast-based screening for therapeutic drugs to be used for treating mitochondrial diseases associated with dominant mutations in the nuclear *ANT1* gene was performed recently. For this purpose, yeast cells with Aac2p adPEO-type mutation Met114Pro corresponding to human ANT1 Leu98Pro mutation were used [94]. More than one thousand compounds were screened for their ability to suppress the defective respiratory growth phenotype of the haploid strain *Δaac1Δaac2* expressing mutant *aac2^Met114Pro^* from the centromeric vector. Five molecules were identified: pergolide mesylate, benzydamine HCl, otilonium bromide, trifluoperazine 2HCl, and sertraline HCl. These drugs were not able to rescue the growth defect of the *Δaac1Δaac2* strain but rescued the oxidative defect of Aac2p mutants Ala128Pro and Ser303Met, corresponding to the pathogenic mutations Ala114Pro and Val289Met. The drugs were effective in reducing mtDNA instability in the heteroallelic *AAC2*/*aac2^Met114Pro^* strain and also in heteroallelic strain carrying the Arg96His mutation equivalent to the dominant to ANT1 Arg80His mutation associated with mitochondrial disease (OMIM 617184) [94]. These results suggest that medical treatment of mitochondrial diseases associated with the dysfunction of ANTs may be developed in future.

## 4. ATP-Mg/P_i_ Carrier

In addition to the equimolar ADP/ATP antiport by AAC, cells also require the transporter capable to mediate the net import of adenine nucleotides into mitochondria to buffer the changes in intramitochondrial ATP concentration in the context of mitochondrial dynamics and to supply mitochondria with the pool of adenine nucleotides in dividing cells [95]. The activity of such a transporter catalysing the import of ATP into mitochondria in exchange for phosphate has been described in mammalian mitochondria. Identification of the corresponding carrier protein is another example of the power of yeast genetics. It resulted from ongoing controversy concerning the viability of mutant strain with deleted *AAC2*. Although the mutants with deleted *AAC2* were described in literature as viable on fermentable carbon sources (see Section 2 and Section 3), there was a considerable number of attempts to prepare the mutant that kept failing. Numerous investigators, thus, considered the *AAC2* gene essential. The resolution to this controversy came when it was found that the tolerance of the *AAC2* deletion is strain specific, and the gene required for survival of *Δaac2* cells—*SAL1* (suppression of *aac*
lethality)—was characterized [96].

Sal1p encodes for a protein that belongs to the calcium-binding domain containing subfamily of the MCF as it contains a calcium-binding regulatory domain, not present in most MCF carriers, at its N-terminus. The protein catalyses import of ATP-Mg^2+^ to mitochondria in exchange for HPO_4_^2−^ (P_i_) and is thus also known as the ATP-Mg/P_i_ carrier [97]. It has to be, however, emphasized here that net import of ATP to mitochondrial matrix can also be performed by the two out of three AAC isoforms in *S. cerevisiae*—Aac2p and Aac3p. This fact rationalizes the synthetic lethality of *Δsal1Δaac2* double mutants, which stood behind the Sal1p identification [96,98,99].

### Human Diseases Associated with ATP-Mg/P_i_ Carrier

In human cells, the pool of adenine nucleotides in mitochondrial matrix is maintained by three calcium-binding isoforms of ATP-Mg/P_i_ carrier—APC1 (SCaMC-1, short calcium-binding mitochondrial carrier, SLC25A24), APC2 (SCaMC3/SLC25A23) and APC3 (SCaMC2/SLC25A25) [100], and the fourth ATP-Mg/P_i_ carrier (SCaMC-3-like/SLC25A41), which lacks N-terminal regulatory domain and is calcium independent [101]. Individual isoforms are expressed in a tissue-specific manner [100,101]. Apc1 has been associated with Fontaine progeroid syndrome (Gorlin–Chaudhry–Moss syndrome, OMIM 612289). Clinical features include craniosynostosis, midface hypoplasia, bone dysplasia, microphthalmia, lipoatrophy, cutis laxa, hypertrichosis, accelerated aging and early demise in certain cases. Autosomal dominant mutations, the single amino acid substitutions, Arg217His or Arg217Cys, in Apc1p were identified in different patients [102,103,104,105]. Fibroblasts from patients exhibit mitochondrial swelling, low ATP content in mitochondrial matrix without reduction of mtDNA copy number and increased sensitivity to oxidative stress [105].

The overexpression of APC1 has been described as the general feature of cancer cells. It appears to promote the cell survival by increased ADP/ATP mediated intra-mitochondrial Ca^2+^ buffering [106]. Further studies are, however, needed to elucidate the role of the APC1 mutations both in Fontaine progeroid syndrome and in the aetiology of cancer.

To study the function of human ATP-Mg/P_i_ carriers, three isoforms, APC1-3, were individually expressed in yeast, but only APC3, not APC1 and APC2, suppressed lethality of *Δaac2Δsal1* cells on medium containing glucose [107]. As expected, none of the human proteins supported respiratory growth on medium-containing glycerol when introduced into *Δaac2*, *SAL1* strain, as human ATP-Mg/P_i_ carriers do not catalyse ADP/ATP exchange in vivo. It should be noted here that the APC1 used in this study differed from the variant for which the transport activity was demonstrated in in vitro assays [100] in the first 54 amino acids. When, in another study, the SCaMC-3-like isoform was expressed in yeast, it was shown to stimulate the uptake of ATP by mitochondria isolated from *Δsal1* strain in the presence of carboxyatractyloside (the inhibitor of AAC) [101].

To study the regulation of ATP-Mg/P_i_ carrier, full-length APC1 or truncated versions without the regulatory domain consisting of two pairs of fused calcium-binding EF-hands but containing different parts of linker domain were overexpressed in yeast, purified and used in studies including functional assays of protein reconstituted in proteoliposomes and thermostability assays. It was proposed that amphipathic α-helix that serves as the linker domain between N-terminal regulatory domain and C-terminal mitochondrial carrier domain becomes mobile upon binding of calcium and could block the transport of substrates [108]. It was demonstrated that calcium regulation of APC1 uses a locking pin mechanism involving the amphipathic α-helix. In the calcium-bound (transport active state), N-terminus of the amphipathic α-helix is bound to a cleft in the regulatory domain. In the calcium-free (transport inactive state), the cleft closes, and the amphipathic α-helix is released and binds to the carrier domain via its C-terminus, causing inhibition of the carrier [108].

Recently, the autosomal dominant inherited missense variant of the APC3 gene was identified in two families with kidney stones. This is the first report linking kidney stones to a dysfunctional mitochondrial carrier. All five patients had a heterozygous dominant mutation manifested as Gln349His substitution. The human wild-type APC3b isoform found in kidneys and the disease variant were expressed in yeast mitochondria, purified, analysed by thermal stability assays and reconstituted in liposomes to measure transport rates [109]. The transport activity of the Gln349His mutant was severely affected and calcium-regulated ATP transport was reduced to ~20% of the wild-type. It is supposed that APC3b Gln349His mutant may augment urine lithogenicity through impaired supply of ATP for solute transport processes in the kidney, and/or for purinergic signalling [109].

## 5. Phosphate Carrier

Inorganic phosphate (P_i_) is translocated into the mitochondria by the phosphate carrier (PiC), another member of mitochondrial carrier family, in yeast encoded by *MIR1* gene [110]. Phosphate carrier exploits the proton gradient generated across the inner mitochondrial membrane by respiration chain and symports P_i_ with protons (H^+^) into the matrix, where it is required, along with other use, for phosphorylation of ADP to ATP by ATP synthase. Analogous to the situation described above for ADP/ATP carrier, *S. cerevisiae* harbours also the second isoform of PiC, named PiC2, which likewise contributes to the import of phosphate to the mitochondria. Double *Δmir1Δpic2* mutant is unable to grow on non-fermentable respiratory substrate (e.g., glycerol) and generates ρ mutants at high frequency (40–50%) [111].

Surptisingly, PiC2 turned out to be bifunctional carrier protein transporting not only phosphate, but also another inorganic substrate—copper. Copper is the intrinsic component of cytochrome *c* oxidase, the terminal oxidase of the mitochondrial respiration chain [112]. *S. cerevisiae Δpic2* deletion mutant displayed very poor growth phenotype on respiratory carbon source (glycerol + lactate) with a mitochondrial matrix targeted copper competitor. PiC2 represents the first mitochondrial copper transporter identified [113].

The Mir1 protein represents the major isoform of mitochondrial phosphate carrier in yeast *S. cerevisiae* when compared to PiC2 with a significantly lower expression level [111]. The closest orthologue of Mir1 in human cells is the mitochondrial phosphate carrier PiC encoded by the SLC25A3 gene. Due to alternative splicing, two isoforms of PiC protein can arise, marked as isoform A and isoform B. While an isoform B is present ubiquitously, although in quite a low amount, the expression of isoform A is tissue-specific with the highest abundance in heart and muscles [114,115]. The same tissue distribution has been reported for bovine isoforms [116]. The different enzymatic properties of the isoforms suggest that while the role of isoform B is to provide mitochondria in all tissues with basal phosphate uptake, the isoform A provides additional transport in the high energy demanding tissues [116].

Mammalian phosphate carrier, but not the yeast one, is synthetized with a N-terminal presequence, which is cleaved off after entering mitochondria [40,110]. Deletion of this presequence is necessary for an effective heterologous import of mammalian PiC (mPiC) into the yeast mitochondria [117]. Although the ablation of presequence is essential for mPiC incorporation, it is still insufficient for the correct function of protein in yeast. The expression of rat cDNAs corresponding to PiC isoform A or isoform B with or without N-terminal presequence (rPiC or ΔNrPiC) has not recovered viability of the yeast *Δmir1* strain on media with glycerol, indicating that there must exist another difference between yeast and mammalian phosphate carrier [118]. The mystery was solved after comparing sequences of randomly mutated ΔNrPiC genes from plasmids isolated from the *Δmir1* revertants with renewed viability on glycerol plates after transformation. Sequence analysis of these mutant ΔNrPiCs revealed that these genes carried a mutation of Phe267Ser or Phe282Ser, which enabled the functional expression of rat PiC homologue in yeast. Both identified phenylalanines were found to be conserved in rat and human PiCs (in both isoform A and isoform B) but not in yeast Mir1p [118]. Consistent with this observation were also results obtained from measurement of transport activity of different versions of rat PiC in *Δmir1* isolated mitochondria. Moreover, experiments with human PiC homologue (full-length; without N-terminal presequence and without N-terminal presequence bearing phenylalanine substitutions) have shown results analogic with that of rat protein in yeast [118]. All these observations have crucial importance for further investigation of human phosphate carrier with the use of yeast model system.

### Human Diseases Associated with Phosphate Carrier

Disorders such as hypertrophic cardiomyopathy and skeletal myopathy, often coupled with lactic acidosis, have been found to be associated with PiC deficiency in humans. The mutation in isoform A leading to the introduction of a novel splicing site (insertion of 8 base pairs, which leads to a frame shift and early stop in the first quarter of the protein) resulting in reduction of a wild-type allele expression (more than 95%) has been identified in the heart and skeletal muscle of certain patients [119]. Another mutation responsible for tissue-specific defect in mitochondrial phosphate carrier was identified previously in two little girls (sisters), none of which lived longer than several months. A single substitution of highly conserved glycine for glutamic acid has been reported in both patients [120]. Glycine at position 72 was found to be conserved not only in mitochondrial phosphate carrier from different species (i.e., human, yeast, frog), but also in some other mitochondrial carriers, such as ADP/ATP carrier, citrate transporter or mitochondrial iron transporter [120]. The effect of Gly72Glu substitution has been tested in yeast double mutant *Δmir1Δpic2*, unable to grow on media with non-fermentable carbon source (glycerol). Expression of human ΔNhPiC with Gly72Glu mutation has not restored growth of *Δmir1Δpic2* on media with glycerol. However, the effect of mutation is difficult to assess in this case, due to mild rescuing effect of wild-type human PiC expressed in *Δmir1Δpic2* cells [120]. This phenotype is probably associated with lack of function of expressed native human PiC in yeast caused by two phenylalanines at specific positions as mentioned above [118]. A variety of other SLC25A3 mutations with pathological effect on phosphate carrier function have been identified to date in both PiC isoforms, many of them still waiting for deeper investigation.

## 6. Pyruvate Carrier

Considering the mitochondria-dependent respiratory metabolism, pyruvate is the key intermediate of energy metabolism, which needs to be imported into the mitochondrial matrix in order to enable the complete oxidation of substrates (e.g., glucose, glycerol) to carbon dioxide and water. Despite obviously central position of mitochondrial pyruvate carrier (MPC) in carbohydrate, lipid and amino acid metabolism [121], its molecular identity remained a conundrum for several long decades. For a time, the carrier of the then unknown function (YIL006w), belonging to mitochondrial carrier family, was thought to transport pyruvate [122] but it was later identified as NAD^+^ transporter [123] (see Section 9). It was only ten years ago that two groups succeeded in identification of the carrier [124,125]. Yeast mitochondrial pyruvate carrier turned out to be a heteromeric complex with components being three previously uncharacterized membrane proteins, Mpc1p, Mpc2p and Mpc3p, having homologues found also in mammals. More recently, active MPC was shown to be heterodimer [126] with alternating composition [127] presumably helping to adapt the yeast energy metabolism to different substrates availability. As already mentioned above, when yeast cells grow under fermentative conditions, significant portion of glucose is only partially oxidised to ethanol (Crabtree effect), and only a small amount of pyruvate is imported into the mitochondria by MPC consisting of Mpc1 and Mpc2. This composition of MPC heterodimer alternates with Mpc1/Mpc3 pair, which ensures the higher pyruvate flux during respiratory growth when substrate (e.g., glycerol) has to be completely oxidised in mitochondria. Mitochondrial pyruvate carrier, thus, participates in the Crabtree effect in baker’s yeast [127]. Deletion of *MPC1*, or simultaneous deletion of *MPC2* and *MPC3*, results in impaired growth on minimal synthetic media without amino acids while no growth defect was observed on rich media, even in a case of triple *Δmpc1Δmpc2Δmpc3* mutant. The growth phenotype of a single *Δmpc2* mutant was milder and no growth defect was observed with *Δmpc3* strain [124,125].

### Human Diseases Associated with Pyruvate Carrier

Human pyruvate carrier Mpc is a heterodimer of Mpc1 and Mpc2 subunits and both subunits complement the phenotypes of deletions of respective paralogues when expressed in yeast [124,125]. Before the identification of genes encoding for MPC, a child patient of healthy consanguineous parents was diagnosed with metabolic disorder including neonatal encephalopathy, severe hyperlacticaemia and microcephaly. Biochemical analysis revealed a defect of mitochondrial pyruvate transport. Progressive neurological deterioration followed, and the child died aged 19 months. Afterwards, her unborn sibling also suffered from defect in mitochondrial pyruvate oxidation and the foetus was aborted [128]. An autosomal recessive metabolic disorder caused by MPC deficiency was later studied in two additional families and samples from the above-mentioned patient. The mutations identified were Leu79His or Arg97Trp in Mpc1. Wild-type and patient-derived Mpc1 mutants were expressed in yeast strain *Δmpc1Δmae1* [124]. This strain manifests a profound growth defect on glucose media due to the absence of Mpc1p and Mae1p, a malic enzyme that converts malate to pyruvate in mitochondrial matrix. The Leu79His Mpc1 mutant suppressed the growth defect significantly less effectively than wild-type Mpc1 and Arg97Trp mutant did not improve the growth at all, which corresponded to the severity of defects observed in human mitochondria.

In a further study, another mutant allele of *MPC1* was characterized in patient’s cells. Due to mis-splicing, *MPC1* C289T results in two protein variants, a truncated version of a protein and a full-length mutant with amino acid substitution Arg97Trp. Truncated Mpc1p fails to stabilize Mpc2p and to form functional MPC complexes. Less stable Arg97Trp mutant, when overexpressed, forms complexes with Mpc2p that retain pyruvate transport activity [129].

## 7. Aspartate-Glutamate, Oxodicarboxylate and Dicarboxylate Carrier

Regardless of whether the preferred pathway of energy metabolism is fermentation or respiration, sustaining the redox homeostasis is absolutely indispensable for cell survive. The malate-aspartate redox shuttle is a cyclic pathway that mediates the import of reduced equivalents from the cytosol into the mitochondria. In the full circle of the process NADH is oxidised to NAD^+^ in the cytosol and electrons are delivered to the mitochondria as part of malate molecule. This Krebs cycle intermediate is then oxidised to oxaloacetate in the matrix while cytosol-derived electrons reduce mitochondrial NAD^+^ to NADH. Among the components of the malate–aspartate shuttle there are two translocators of the mitochondrial carrier family, the amino acids transporting aspartate–glutamate carrier (AGC) [130] and oxodicarboxylate carrier (ODC) [131].

Actually, the identification of AGC in *S. cerevisiae*—*AGC1*—provided a strong evidence for the existence of functional malate–aspartate redox shuttle in yeast as it had been questioned due to the absence of proton pumping complex I of respiratory chain [132] and the presence of NADH dehydrogenase on the cytosolic side of the inner mitochondrial membrane [130]. Interestingly, in this case, the identification of yeast carrier followed the identification and characterisation of human orthologues [130]. Yeast AGC can function both as uniporter and antiporter. Under uniport mode of action it imports glutamate into the matrix for nitrogen metabolism and biosynthesis of ornithine (see Section 12 and Figure 1). As antiporter, AGC is required for the operation of the malate-aspartate shuttle, essential for yeast growth on acetate and fatty acids as carbon sources [130]. In this mode, it exports aspartate, an acidic amino acid, in exchange for another acidic amino acid, glutamate. Import of glutamate is accompanied by the translocation of proton in the same direction, utilizing a mitochondrial inner membrane proton gradient as driving force. The amino group of aspartate is in the cytosol transferred to oxoglutarate in a transamination reaction, giving rise to oxaloacetate and glutamate.

Cytosol is continuously supplied by oxoglutarate, an acceptor of amino group, by the second carrier involved in malate-aspartate NADH shuttle, ODC. Given its strict counter exchange mechanism, export of oxoglutarate goes together with the import of the equimolar amount of malate, generated in the cytosol by the reduction of oxaloacetate, to the matrix [131,133]. There are two isoforms of ODC in *S. cerevisiae*. ODC1, the more abundant isoform, is the subject to the repression by fermentable carbon sources (glucose, galactose), while a relative amount of ODC2 is under such conditions even higher as compared to the situation on respiratory substrates (glycerol, ethanol and lactate). Double *Δodc1Δodc2* mutant grows both on non-fermentable and fermentable carbon sources, indicating that ODC carrier is non-essential for yeast cell respiration [131]. The oxodicarboxylate carrier is utilized also for the transport of 2-oxoadipate, intermediate of lysine metabolism, across the inner mitochondrial membrane.

The oxodicarboxylate carrier is not the only member of mitochondrial carrier family involved in the transport of malate into the mitochondria in *S. cerevisiae*. The interplay between the cytosolically localised glyoxylate cycle, allowing the utilisation of C2 carbon sources (ethanol, acetate) and lipids for the biosynthesis of saccharides, with Krebs cycle taking place in the mitochondrial matrix is allowed by the dicarboxylate carrier (DIC) [134,135]. Among the predominant substrates of this carrier are both succinate, Krebs cycle intermediate and glyoxylate cycle product, and malate, intermediate in both cycles [133]. Catalysing the electroneutral exchange of dicarboxylates by an antiport mechanism DIC imports malate and succinate into the mitochondria and exports inorganic phosphate. Thus, in analogy to the situation described above for the Sal1p, DIC utilizes the intramitochondrial pool of P_i_ generated by another already introduced member of mitochondrial carrier family, the phosphate carrier.

### 7.1. Human Diseases Associated with Aspartate-Glutamate Carrier

Two isoforms of the AGC exist in human cells, AGC1 (named aralar1, encoded by SLC25A12) and AGC2 (named citrin, encoded by SLC25A13) [136]. Both AGC isoforms belong to the subfamily of MCF whose activity is regulated by extramitochondrial Ca^2+^. This is associated with the presence of N-terminal domain containing eight EF-hand Ca^2+^ binding motifs, which are missing in the yeast Agc1p [130,137]. The physiological role of the AGC is to supply cytosol with mitochondrially synthesized aspartate in exchange for glutamate and proton from the cytosol, in an electrogenic process. As in yeast, human AGC, together with oxoglutarate carrier [138,139,140,141], plays a crucial role in the transport of NADH-reducing equivalents from the cytosol to the mitochondrial matrix as being a part of the malate-aspartate shuttle [136] and the proper cooperation of these two mitochondrial carriers is necessary for maintaining redox potential inside the cell.

The expression profile of AGC1 and AGC2 throughout the human body is not uniform, and each isoform is dominant in a certain tissue. While AGC1 dominates in the brain, heart and skeletal muscle, the highest expression level of AGC2 is observed primarily in the liver and kidney [142,143,144].

Mutations in the SLC25A12 gene causing defects in the brain-dominant isoform AGC1 lead to rare neurological disease (OMIM 612949) connected with global cerebral hypomyelination, developmental delay, epilepsy and hypotonia [145,146]. Moreover, it seems that there is a certain link between impaired AGC1 function and autism [147]. Since the impaired function of AGC1 naturally leads to the dysfunction of malate-aspartate shuttle and reduced effectivity of saccharide metabolism, the above-mentioned disorders are not so surprising. An interesting neuroprotective effect has been observed in experiments with *AGC1* knockout mice pups treated with injections of β-hydroxybutyrate (βOHB), the main ketone body produced in ketogenic diet. Supplementation with βOHB was shown to be highly effective in rescuing basal and agonist-stimulated mitochondrial respiration in *AGC1* knock-out neurons and was able to compensate for the metabolic failure caused by AGC1/malate-aspartate shuttle deficiency [148,149].

Mutations in the SLC25A13 gene encoding AGC2 are more common than mutations in AGC1, with the highest prevalence in Japanese population (an estimated 1 in 100,000 to 230,000 individuals) and are relatively frequent in East Asia and the Middle East population [150]. Diseases caused by AGC2 mutations are collectively called citrin deficiency, and their common feature is an impaired urea cycle associated with a failure in the export of aspartate from the mitochondria. There are recognized two age-dependent disease phenotypes—neonatal intrahepatic cholestasis (NiCCD, OMIM 605814) in newborns and the adult-onset type 2 citrullinaemia (CTLN2, OMIM 603471) with an onset at the age of 11–79. NiCCD is a metabolic disorder characterized by prolonged jaundice, poor growth, intrahepatic cholestasis, hypoglycaemia, hypoproteinaemia and increased serum citrulline. Interestingly, most patients show spontaneous health improvement during the first year of life, but in certain cases, they may still suffer from continued failure to thrive and dyslipidaemia caused by citrin deficiency (FTTDCD) and may develop CTLN2 later in life [151,152]. CTLN2 shares similar characteristic markers as NiCCD including hypoproteinaemia, hyperammonaemia, citrullinaemia, but with more severe clinical symptoms such as neuropsychiatric disorders, pancreatitis, hepatic steatosis, coma and brain oedema. Effective treatment of CTNL2 often involves liver transplantation, without which the disease is usually fatal. A special diet is typically insufficient [152].

From more than 90 mutations identified to date in patients suffering from different types of citrin deficiency, the effect of the significant number has been investigated by utilizing yeast *S. cerevisiae*. It has been shown previously that human AGC isoforms can functionally complement the loss of yeast Agc1p and rescue *Δagc1* mutant cells unable to grow on acetate and oleic acid [130]. Functional analyses of different mutant variants of human AGC2 expressed in yeast have either confirmed the deleterious effect of the specific mutation on a carrier biological activity or has given proof about its harmlessness [153]. For example, variants bearing one of the following mutations, Gly437Glu, Leu598Arg, and Glu601Lys, were unable to restore the growth of *Δagc1* cells, while another variant Thr546Met was able to restore yeast growth to approximately 25% of the wild-type level. The alternative variant Pro632Leu, found not to be associated with AGC2-deficiency disorders, has proven to be fully functional in *Δagc1* yeast growth recovery [153]. An interesting finding was obtained when a truncated variant of the human AGC2 transporter was investigated. This truncation involved deletion of the first 34 amino acids from the protein N-terminus including the portion of the first EF hand motif (EF1). The markedly changed structure of the protein was presumably the reason why this AGC2 variant was not able to restore normal growth of *Δagc1* strain, since variant with two substitutions in EF1 (Glu28Ala and Asn30Ala) was still able to restore the *Δagc1* strain viability to approximately 50% of the wild type. Additionally, the intracellular localisation of the truncated AGC2 variant was also altered, matching with the loss of function [153]. The effect of the other mutation identified in a patient with NiCCD was also successfully examined in a yeast model system [154].

### 7.2. Human Diseases Associated with Oxodicarboxylate Carrier

Human ODC (SLC25A21) exchanges mainly 2-oxoadipate or 2-aminoadipate for 2-oxoglutarate by a counter-exchange mechanism and plays a central role in the catabolism of lysine, hydroxylysine and tryptophan. The identification of human ODC was achieved owing to the characterisation of two yeast isoforms, Odc1p and Odc2p [131] and sequences of ODC orthologues in *Caenorhabditis elegans* and *Drosophila melanogaster* [155]. Deficiency of ODC is a cause of mtDNA depletion, mitochondrial dysfunction and the accumulation of oxoadipate and quinolinic acid, which results in toxicity in spinal motor neurons leading to a spinal muscular atrophy-like disease. Mitochondria of the patient carrying the homozygous mutation in the SLC25A21 allele, identified as Odc1 Lys232Arg, were unable to import 2-oxoadipate into the matrix [156]. The yeast strain *Δyhm2Δodc1Δodc2* was prepared to investigate the roles of mitochondrial carriers Yhm2p [157], Odc1p and Odc2p [131] in the assimilation of nitrogen and in the biosynthesis of lysine. This triple mutant showed markedly reduced growth on glucose medium with ammonia as the main nitrogen source, which was rescued by glutamate or individual expression of *YHM2*, *ODC1* or *ODC2*. The triple mutant also exhibited a lysine auxotrophy that could be partially rescued by the addition of 2-aminoadipate [158]. Only the concurrent absence of Yhm2p, Odc1p or Odc2p impairs the export of 2-oxoglutarate and 2-oxoadipate from the mitochondrial matrix to cytosol where these metabolites are necessary for the synthesis of glutamate and ammonium fixation or for lysine biosynthesis, respectively. The *Δyhm2Δodc1Δodc2* strain thus appears to be suitable for the investigation of the pathogenic potential of mutations in human [158].

### 7.3. Human Diseases Associated with Dicarboxylate Carrier

As in yeast, human DIC (SLC25A10) transports dicarboxylates and phosphate across the inner mitochondrial membrane [135,159]. A recessive mutation in the DIC gene of a patient creating a premature stop codon, synonymous and an intronic mutations were associated with reduced RNA quantity, aberrant RNA splicing and absence of DIC protein. A patient without a functional DIC protein has a mitochondrial disorder with severe epileptic and progressive encephalopathy with complex I deficiency [160]. Yeast *S. cerevisiae* was used to investigate the cellular phenotype caused by the absence of DIC functional homologue Dic1p. Deletion strain *Δdic1* [161] showed reduced efficiency of mitochondrial respiration and a decrease in mtDNA copy number, similar to what has been observed in the patient’s skeletal muscle. Cells without DIC are prone to oxidative stress damage since the patient’s fibroblasts were depleted in the main antioxidant molecules NADPH and GSH. Similarly, yeast *Δdic1* strain displays growth defect in respiratory medium after addition of H_2_O_2_ [160].

## 8. Citrate Carrier

In addition to ODC and DIC, the third route for the cytosolic malate to enter the mitochondria is represented by the citrate carrier (CTP, CIC or tricarboxylate carrier), in *S. cerevisiae* encoded by *CTP1* [162]. The import of malate is, in this case, coupled with the export of another Krebs cycle intermediate, citrate. In fact, yeast CTP manifests stricter substrate specificity for tricarboxylates (citrate, isocitrate) than the citrate carrier in mammals. Citrate exported from mitochondria serves as a substrate for the anabolic reactions taking place in the cytosol and leading to the biosynthesis of fatty acids and sterols. In cancer cells of multicellular organisms, citrate fuels the cell metabolism and proliferation [163,164,165].

### Human Diseases Associated with Citrate Carrier

Impaired function of mitochondrial citrate carrier (CIC) in humans is always linked with decreased level of citrate and isocitrate in urine in contrast to elevated level of other Krebs cycle intermediates, including malate, succinate, fumarate or α-ketoglutarate, and its derivates l-/d-2-hydroxyglutarate. While an elevated level of l-2-hydroxyglutarate or d-2-hydroxyglutarate may be a biochemical marker of defects in different enzymes, high level of both isomeric compounds together is typical for hydroxyglutaric aciduria specifically caused by citrate carrier dysfunction [166]. Combined d-2-/l-2-hydroxyglutaric aciduria is characterised by serious brain abnormalities that become evident early after birth. Affected infants usually suffer from severe epileptic seizures, hypotonia, problems with breathing and feeding and often die at the age of few months rarely years.

More than twenty mutations in SLC25A1 gene encoding human citrate carrier have been identified to date (see [167] for review). Clinical manifestation of the CIC impairment partly depends on the position of the mutation in carrier structure and ranges from mild to serious symptoms [167]. Individual mutations have been investigated by various approaches, and the harmfulness of several SLC25A1 pathogenic variants was examined also in yeast. Two specific mutations, Gly130Asp and Arg282His, localised in CIC transmembrane helixes 3 and 5 were connected with the absence of the *corpus callosum* in the patient suffering from already mentioned health problems [168]. Both amino acid residues were found to be highly conserved among species. An in silico modelling clearly indicated a deleterious effect of the mutations on substrate binding ability and an entire transporting function of the citrate carrier. This was indeed confirmed by the defective growth of a yeast strain harbouring the relevant mutations inside its genome (Ctp1 Gly117Asp + Arg276His) cultivated upon stress conditions (elevated temperature and presence of hydroxyurea in cultivation medium). The observed growth phenotype of the double mutant was comparable with the *CTP1* deletion mutant missing the whole transporter. Kinetic studies with reconstituted protein Ctp1 harbouring only one of the mutations showed complete loss of transport function in case of arginine to histidine substitution and significant decrease in transport activity when glycine was changed for aspartic acid [168].

## 9. NAD^+^ Carrier

Despite the functionality of malate–aspartate NADH shuttle in yeast [130], cells also require the mechanism responsible for the generation and sustaining of the intramitochondrial NAD^+^ pool as this mitochondrial energy metabolism fuelling cofactor is synthesized in the cytosol in *S. cerevisiae*. This situation is thus conceptually identical to the situation already described for the AAC and Sal1p carriers in the context of intramitochondrial ATP pool buffering. Although first suggested to be the pyruvate carrier [122], identity of *YIL006w* gene product was soon after re-evaluated and it was identified as mitochondrial NAD^+^ carrier (Ndt1p), together with its paralogue (Ndt2p) [123]. Both transport proteins belong to the mitochondrial carrier family [133]. Ndt1p carrier transports NAD^+^ cofactor, to a lesser extend (d)AMP and (d)GMP nucleotides, but practically no NADP^+^, NADH or NADPH. Mutants lacking either of the paralogues have reduced levels of intramitochondrial NAD^+^ and NADH and reduced activities of mitochondrial enzymes requiring oxidised form of this cofactor, pyruvate dehydrogenase (PDH) and acetaldehyde dehydrogenase (ACDH). Double *Δndt1Δndt2* mutation has even more profound effect on mitochondrial NAD^+^ and NADH content and the growth of these double mutants on non-fermentable carbon sources (ethanol, lactate, pyruvate, acetate) is characterized by pronounced lag phase, especially on ethanol [123]. Import of NAD^+^ into the matrix is probably linked with the transport of (d)AMP/(d)GMP nucleotides in opposite direction.

Only recently, the carrier for NAD^+^ was identified in humans as proteins encoded by gene SLC25A51 (MCART1) and an almost identical, but not widely expressed, paralogue SLC25A52 (MCART2) [169,170,171]. When expressed in the *Δndt1Δndt2* yeast strain, both can restore the growth on minimal media with non-fermentable carbon sources and SLC25A51 was also shown to normalize the intramitochondrial NAD^+^ level, indicating that they can substitute for the function of yeast carriers [169,170,171]. Moreover, when the uptake of NAD^+^ by mitochondria isolated from strains expressing human proteins was measured, it was shown that both human proteins can support the transport of NAD^+^, which is completely absent in *Δndt1Δndt2* mitochondria [171]. Conversely, the expression of yeast *NDT1* can rescue the respiration defects observed in SLC25A51-knock out lines [169,170].

## 10. Thiamine Pyrophosphate Carrier

Five enzymes in yeast require the thiamine pyrophosphate (ThPP) as a cofactor for their function. Two of them—pyruvate decarboxylase (PDC) and transketolase—are located in the cytosol, whereas the other three—acetolactate synthase (ALS) and the E1 components of pyruvate dehydrogenase and oxoglutarate dehydrogenase are—in the mitochondrial matrix. Demand for the ThPP in both cell compartments and the fact that thiamine pyrophosphokinase, an enzyme responsible for ThPP synthesis, is only localised in the cytosol [172,173] supported the search for mitochondrial ThPP transporter. It was shown that the carrier protein effectively transporting ThPP in *S. cerevisiae* either by itself or in exchange for thiamine monophosphate (ThMP), is protein Tpc1p encoded by *YGR096w* gene [174]. The carrier was identified when the gene was overexpressed in bacteria and purified protein was reconstituted into artificial liposomes. Measurements with radioactively labelled dATP and proteoliposomes pre-loaded with ThPP or ThMP (or with other substrates) had given a proof about a substrate specificity and function of yeast Tpc1p [174]. Viability of *TPC1* knockout strain on complete media containing either fermentable or non-fermentable carbon source is comparable with wild-type strain, whereas *Δtpc1* cells require thiamine or branched-chain amino acids for growth on minimal media with fermentable carbon sources [174]. Consistently, the activity of acetolactate synthase, which catalyses the first step in the synthesis of the branched-chain amino acids (valine, leucine and isoleucine) and needs ThPP as cofactor, is decreased in *Δtpc1* cells [174,175]. The ability of branched-chain amino acids to restore the growth indicates that ThPP is in yeast mitochondria mainly required for the biosynthesis of these amino acids. Surprisingly, no growth defect nor the differences in the activity of mitochondrial ThPP-requiring enzymes was observed when *Δtpc1* strain was cultivated in the media containing ethanol or other non-fermentative substrate. This may indicate that under these conditions another transporter, yet to be identified, can supply mitochondria with ThPP [174].

The closest human relative of Tpc1p, with the sequential homology of about 28% at the level of amino acids, is protein TPC (encoded by SLC25A19), identified previously as the deoxynucleotide carrier (DNC) [176,177]. Although both proteins are able to mediate ThPP transport across the IMM, their function characteristics differ in few aspects. While yeast Tpc1 can function as an antiporter and upon specific conditions as a uniporter, human TPC catalyses only exchange and not uniport of substrates [176]. These homologues also have different affinity and transporting ability to secondary substrates (other than ThPP and TmPP) including (deoxy)nucleotides, nucleosides and purines, as measured in vitro [174,176].

### Human Diseases Associated with Thiamine Pyrophosphate Carrier

Several serious diseases are associated with mutations in the SLC25A19 gene. One of them, linked with a substitution Gly177Ala that results in dysfunction of TPC1 protein is Amish microcephaly (OMIM 607196). The disease is characterized by severe congenital microcephaly, with no brain and motor development, highly elevated α-ketoglutarate, lactic acidosis and premature death [178,179]. The increased level of α-ketoglutarate is probably due to impaired function of α-ketoglutarate dehydrogenase, while lactic acidosis is the result of reduced pyruvate dehydrogenase activity, both enzymes requiring the cofactor ThPP. Correspondingly, an in vitro exchange assay with mutant human protein containing the Gly177Ala mutation showed reduced ThPP and ThMP transport by 70% compared with the wild-type protein [177].

Another substitution in the SLC25A19 gene product, Gly125Ser, was identified in patients suffering of bilateral striatal necrosis (OMIM 613710) [180]. This neurological disorder is clinically manifested by ataxia, hypotonia, hyporeflexia, transient limb paresis, or fixed dystonic posturing, and by a disorder of consciousness [181,182]. To test the significance of the above-mentioned mutation, plasmids containing the coding sequence of the normal human SLC25A19 gene or the mutant version of a gene were introduced into the yeast *Δtpc1* strain. Expression of the wild-type SLC25A19 fully complemented the thiamine auxotrophy of the *Δtpc1* mutant unable to grow on media without thiamine, while Gly125Ser mutant complemented thiamine auxotrophy only partially [180]. The observed phenotype confirmed that the effect of mutation on the activity of human TPC protein is not negligible.

Several other mutations were identified in the human SLC25A19 gene with less deleterious effect on TPC1 functionality, probably due to the position of substitution in the protein primary structure, not influencing the transferring ability of the ThPP transporter significantly [183].

## 11. S-Adenosylmethionine Carrier

Mitochondrial DNA, as numerous other information storing macromolecules, undergoes different types of modifications. One such chemical modification is methylation, requiring the presence of methyl group (–CH_3_). However, universal methyl donor S-adenosylmethionine (SAM) is in *S. cerevisiae* synthesized from methionine and ATP in the cytosol [184], making the existence of SAM transporter inevitable for proper mitochondrial functioning. The carrier for S-adenosylmethionine (Sam5p) was discovered a year after the identification of thiamine pyrophosphate transporter by the same research group and also belongs to the MCF [185]. Capable of both unidirectional transport and antiport of SAM, its physiological function is supplementation of the mitochondrial matrix with this methyl donor and export of SAM demethylation product—S-adenosylhomocysteine [133].

Deletion mutant *Δsam5* is auxotrophic for biotin when grown on fermentable carbon sources (glucose and galactose) and does not grow at all on non-fermentable substrates (glycerol, lactate, ethanol, acetate, and pyruvate) [185]. Both phenotypes are connected to the biosynthetic pathways the SAM is indispensable for. Shortage of S-adenosylmethionine in the mitochondria interferes with adequate intramitochondrial synthesis of biotin, where SAM is essential for the conversion of dethiobiotin to biotin by biotin synthetase (Bio2p) [185,186]. On the other hand, SAM-dependent intramitochondrial synthesis of lipoic acid contributes to the *petite* phenotype of *Δsam5* strain as this methyl group donor is the crucial cofactor of the enzyme catalysing the last step of the biosynthetic pathway—lipoate synthetase (Lip5p) [185,187]. Lipoate is necessary for the activities of PDH and OGDH, both enzymatic complexes involved in utilisation of respiratory, non-fermentable carbon sources. Noteworthy, both phenotypes of the *Δsam5* mutant were suppressed by the artificial expression of SAM synthetase in the mitochondria [185].

### Human Diseases Associated with S-Adenosylmethionine Carrier

In humans, mitochondrial SAM carrier is encoded by a single gene (SLC25A26) that is expressed in all tissues tested and likely represents the only way for import of SAM into mitochondria [188]. Unlike its yeast homologue, mammalian carrier only acts as an antiporter [188]. Mutations in SAM carrier were identified in three unrelated children patients manifesting different symptoms, including acute episodes of cardiopulmonary failure, respiratory insufficiency, lactic acidosis, and progressive muscle weakness. Their mitochondria exhibited reduced intra-mitochondrial methylation of RNA and proteins, reduced translation and diminished levels of coenzyme Q and lipoic acid. Three of the identified point mutations resulted in single amino acid substitutions: Val148Gly in homozygous, and Ala102Val and Pro199Leu in heterozygous patient, and the fourth mutation, homozygous, led to a splice defect resulting in a protein lacking N-terminal 88 amino acid (SAMCD1–88) [189].

Activities of mutant forms of SAM carrier were also tested in yeast as the ability to rescue the growth defect of *Δsam5* strain on non-fermentable carbon source. While expression of wild-type human SLC25A26 rescued the growth to the level comparable with wild-type yeast control, of all tested mutants only Val148Gly was able to partially support cell growth. The remaining mutants were unable to grow to support the growth. Corresponding results were obtained when the transport capacity of individual mutant proteins reconstituted in proteoliposomes was measured. Dramatically reduced transport activity was detected with Val148Gly, and other mutant proteins were completely inactive [189]. Notably, the severity of symptoms related to the nature and activity of mutant proteins, with the severity of symptoms increasing in following order—homozygous Val148Gly, heterozygous Ala102Val and Pro199Leu, and homozygous SAMCD1–88, indicating that ability of mutant alleles to complement yeast grow defect may be relevant indicator.

## 12. Ornithine/(Citrulline) Carrier

Still, another transporter of the eukaryote specific mitochondrial carrier family is involved in the compartmentalised metabolism of nitrogen in eukaryotic cells. The need for the fluent flow of nitrogen containing ornithine, lysine and arginine between cytosol and mitochondria is mediated by carriers for ornithine and related amino acids [190]. When identified in *S. cerevisiae* it was originally assigned a designation Arg11p to reflect its role in the biosynthesis of arginine [191], later a designation Ort1p became widely accepted. Reconstituted in proteoliposomes, Ort1p shows highest substrate specificity for ornithine, followed by arginine and lysine. Citrulline, an intermediate of the urea cycle, is not transported, which is the distinguishing feature of yeast ornithine carrier when compared with human (ORC1, ORC2, ORC3) and plant homologues (BAC1, BAC2) [190,191]. Absence of urea cycle in *S. cerevisiae* is rationalised by the removal of excessive nitrogen directly in the form of ammonia [190].

The biosynthesis of arginine in *S. cerevisiae* involves intramitochondrial formation of ornithine from imported glutamate and subsequent conversion of ornithine to arginine in the cytosol [192]. Before the identity of the ornithine carrier was uncovered, yeast mutants in *ARG11* gene were known to grow poorly in the absence of arginine [193] and one particular mutant (MG409/Arg11-1) was characterised by the increase in cytosolic glutamate pool associated with decreased arginine and ornithine pools [194]. Consistent with these observations, the physiological role of Ort1p is the export of mitochondrially synthesised ornithine to the cytosol in exchange for imported cationic amino acids arginine and lysine, or for protons [133,191]. This is the way in which the ornithine carrier is involved in the biosynthesis of arginine, but also of polyamines, for which arginine is the precursor, in yeast [190].

### Human Diseases Associated with Ornithine/Citrulline Carrier

In humans, SLC25A15 (Orc1, Ornt1) is the ornithine/citrulline carrier associated with hyperornithinaemia–hyperammonaemia–homocitrullinuria (HHH) syndrome, a rare autosomal recessive disorder of the urea cycle (OMIM 238970). Orc1 gene was identified using known sequences of ornithine carriers Ort1p (Arg11p) and Arg13 from *S. cerevisiae* and *Neurospora crassa*, respectively [195]. ORC1 Phe188Δ mutant, caused by 3-bp in-frame deletion, is common in French-Canadian patients with HHH syndrome and encodes an unstable protein. It was investigated by expression in yeast *Δort1* strain [196]. Wild-type human Orc1p was correctly inserted into the yeast inner mitochondrial membrane, exhibited strong antiport activity and displayed stereospecifity for l-ornithine. The Phe188Δ mutant was incorporated significantly less efficiently into the membrane, had reduced uptake of l-ornithine, and lost transport stereospecifity.

Orc1p homozygous mutation Ala15Val was identified in a child patient with childhood onset of HHH syndrome, and its effect on protein activity was tested by complementation of the yeast *Δort1* null mutant which does not grow on minimal medium without arginine [197]. Wild-type Orc1p expressed from multicopy vector restored growth of the cells. Orc1p Ala15Val mutant was present in mitochondria, but it did not support growth of the cells. Abolished transport of ornithine was confirmed also with purified Orc1p Ala15Val mutant reconstituted in liposomes. Upon information about structure and mechanism of mitochondrial carriers [198], it was suggested that Ala15Val mutation present in the first transmembrane helix adjacent to a highly conserved glycine-rich region involved in opening and closing of the Orc1p on the cytosolic side, impairs conformational changes occurring in the catalytic cycle of the carrier [197].

*S. cerevisiae* was used to study the role of SLC25A2 (Orc2, Ornt2) and the function of Orc1 missense mutations [199]. Orc2p, which has the 88% identity with Orc1p, was suggested to act as a second mitochondrial ornithine carrier, in part responsible for the milder phenotype in HHH patients [200]. Surprisingly, Orc2 expressed from a multi-copy vector does not complement the growth of the *Δort1* strain in contrast to Orc1p. Thus, three specific residues (Met72, Ala123 and Gln179) of Orc2p were replaced by the corresponding residues of Orc1 (Ile72, Val123, Arg179) that are highly conserved among Orc1 orthologues. Of the tested single substitutions, the growth of the *Δort1* strain was only rescued by the Orc2p Gln179Arg. The growth was further improved by combined substitutions Gln179Arg/Met72Ile or Gln179Arg/Ala123Val. The triple mutant Orc2p Gln179Arg/Met72Ile/Ala123Val supported growth similarly as the wild-type Orc1p. These experiments in yeast are in concordance with characterisation of purified Orc1p and Orc2p in liposomes. Residue at position 179 is present in substrate binding site at contact point II of Orc1p (Arg179) and Orc2p (Gln179) with Glu180 and is essential for substrate specificity. Orc2p Gln179Arg and the wild-type Orc1p transport only the l-forms of ornithine, arginine, and lysine but wild-type Orc2p and Orc1p Arg179Gln transport also d-forms of these amino acids. Wild-type Orc1p has a V_max_ value of ornithine transport 2.5 times higher than wild-type Orc2p and Orc2p Gln179Arg displays V_max_ value 33 times higher than wild-type Orc2p [201].

A series of Orc1p mutations identified in HHH patients (Met37Arg, Ala70Leu, Leu71Gln, Gly113Cys, Glu180Lys, Phe188Leu, Gln246Lys, Thr272Ile) was also tested by expression in *Δort1* yeast strain. All mutations with exception of Gln246Lys reduced or abrogated the ability of Orc1p to support growth of the cells. Only the Met37Arg and Leu71Gln Orc1 mutants partially restored growth ability [199].

Effects of Orc1p missense mutations causing HHH syndrome Gly27Arg, Met37Arg, Phe188Leu, Arg275Gln and experimentally created mutations Asn74Ala, Phe188Tyr, Ser200Lys and Arg275Lys was tested by complementation of *Δort1* mutant in arginine-less synthetic medium together with transport assays in reconstituted liposomes [202]. Cells expressing Orc1 Asn74Ala or Ser200Lys grew similarly as control cells with wild-type Orc1, but the expression of Orc1 variants Gly27Arg, Met37Arg, Phe188Leu, Phe188Tyr, Arg275Gln and Arg275Lys did not restore cell growth. All Orc1 variants were present in mitochondria in similar amounts.

SLC25A29 (Orc3, ORNT3) is the human mitochondrial carrier of basic amino acids, transporting arginine, lysine, and to a lesser extent ornithine and histidine [203]. Orc3p expression is significantly elevated in many cancer cells and increased import of arginine into mitochondria raised mitochondria-derived nitric oxide levels, what facilitate cancer progression [204]. The Orc3 carrier can rescue the deficient ornithine metabolism in fibroblasts of patients with HHH syndrome [205]. Expression of wild-type carrier Orc1, Orc2 and closely related Orc3 from multicopy vector in yeast *Δort1* yeast strain was further investigated [202]. Only Orc1p rescued growth of the cells, but this may be related to the evidently lower amounts of Orc2p and Orc3p detected in the mitochondria.

## 13. Carnitine Carrier

In yeast, the degradation of fatty acids occurs exclusively in peroxisomes [206]. Fatty acids are oxidised by β-oxidation and acetyl-CoA is produced. In order to complete the oxidation fully to carbon dioxide and water, acetyl-coenzyme A has to be transported to the mitochondria, to feed the Krebs cycle and respiratory chain. Transport of acetyl-CoA from peroxisomes to the mitochondrial matrix can be accomplished by two independent pathways. Acetyl-CoA can either be converted to succinate in glyoxylate cycle in peroxisomes and succinate be transported to the mitochondria, e.g., by DIC, or by the second pathway, in which it is converted to acetylcarnitine by carnitine acetyltransferase and transported to mitochondria by specific carrier. In mitochondria, the acetyl group from acylcarnitine is transferred to CoA by matrix-localised carnitine acetyltransferase, regenerating carnitine and acetyl-CoA, such that the outcome of the entire pathway is the transport of acetyl-CoA to mitochondria (Figure 1).

The carrier that catalyses the transport of acylcarnitine in exchange for carnitine is in yeast encoded by *CRC1* and was identified by two independent groups. One used the strain, in which gene encoding for peroxisomal citrate synthase (*CIT2*) was deleted, such that acetyl-CoA could only be transported to the mitochondria by a pathway that rely on acylcarnitine/carnitine carrier. Mutants generated from this strain were screened for those that would not grow on oleic acid as the sole carbon source. In one of identified complementation groups, the mutations in gene encoding for acylcarnitine/carnitine carrier disabled the transport of acetyl-CoA [207]. The other group identified *CRC1* on the base of homology with genes encoding for acylcarnitine/carnitine carrier in human and rat, overexpressed the gene, and characterised the carrier biochemically [208].

### Human Diseases Associated with Carnitine Carrier

In mammals, including humans, only the oxidation of very long chain fatty acids occurs in peroxisomes while oxidation of shorter fatty acids, still including those with long chain, is localised in the mitochondria. These fatty acids are transported to the mitochondria as acyl esters of carnitine by acylcarnitine/carnitine carrier (CACT, CAC, SLC25A20). As the mitochondrial oxidation of fatty acid provides a main source of energy during fasting as well as in certain tissues, e.g., cardiac muscle or skeletal muscles during exercise, defects in mitochondrial oxidation of fatty acids causes severe disorders that are generally manifested as acute, potentially life-threatening episodes of hypoketotic hypoglycaemic coma induced by fasting. These defects include dysfunction of either any of sixteen enzymes involved in the process itself or defect in acylcarnitine/carnitine carrier. In fact, the acylcarnitine/carnitine carrier was the first mitochondrial carrier associated with human disease [209].

Several mutations in CACT were characterized in patients, some of them have been studied using yeast. When expressed in yeast *Δcic2Δcrc1* mutant, the wild-type human carrier recovers the growth on media containing oleate as a sole carbon source [210]. Two of the mutations associated with different phenotypical manifestation were expressed in same mutant strain. The substitution resulting in Gly81Arg was found to result in inactive protein, as expression of mutant version both did not recover the growth on oleate media, nor did it enable the isolated mitochondria to oxidise fatty acids measured as production of carbon dioxide. The other studied mutant, in which the mutation caused the splice defect, resulting in the production of a protein with a 21-amino-acid extension, failed to recover the growth on oleate but was able to support low level of carbon dioxide production. When amounts of protein detected in yeast mitochondria by Western blot was considered, experiments showed that while the aberrant protein produced by the expression of second mutant retained its activity, the phenotype was caused by low level of functional protein in the membrane [210].

## 14. Other Yeast Mitochondrial Carriers

The genome of *S. cerevisiae* contains 35 open reading frames that encode for the proteins of MCF. In addition to the ones described above, carriers specific for the transport of coenzyme A [211], succinate/fumarate [212], citrate/oxoglutarate [157], oxaloacetate [213], glycine [214], FAD [215], GDP/GTP [216], pyrimidine nucleotides [217], APS and PAPS [218], pyridoxal 5′-phosphate [219], Fe^2+^ [220] and Mg^2+^ [221] have been described (Table 1). Despite the homology with mitochondrial carriers, another two are a protein of the outer mitochondrial membrane that plays a role in fusion of mitochondria [222] and the ATP/AMP transporter localised in the membrane of peroxisomes [223,224]. The last one is a putative mitochondrial carrier that remains to be characterized.

## 15. Conclusions

Mitochondria play crucial roles in both energetic and synthetic metabolism of eukaryotic cells. Their function depends on strictly selective transport of numerous metabolites, many of which are transported by the protein carriers of the MCF. As mitochondrial metabolism, including the transport by mitochondrial carriers, is well conserved between mammals and yeast, the advantages arising from robustness of yeast and the powerful battery of yeast techniques make this single cell eukaryote an excellent model for studying mitochondrial carriers. Yeast tolerates the single deletion of any MCF member and many of deletion mutants have distinct phenotypes. If not, phenotypes can often be detected in the specific strain, e.g., carrying an additional mutation that disables a parallel pathway that may obscure the actual phenotype. As such, yeast is well suited to study the functionality of carriers, in many cases by simply detecting the corresponding phenotypes. This gives us a tool that can be exploited for innumerable applications in the way that would not be possible in any other experimental system. Mutant versions of human mitochondrial carriers associated with genetic diseases can be studied in yeast either by direct expression, by expression of chimeric proteins, or by expressing modified versions of yeast proteins with mutations in corresponding residues. Yeast can also be used in different high throughput screens for chemicals affecting the function of yeast native or expressed human carriers. Here, several of these are described, but indeed many must have been left out as they reach beyond the scope of this review. We believe that the capacity of yeast as of model in the field of mitochondrial transport has not nearly been exhausted yet, and we expect that by learning from yeast we will find out a lot about ourselves.

## Figures and Tables

**Figure 1 microorganisms-09-02044-f001:**
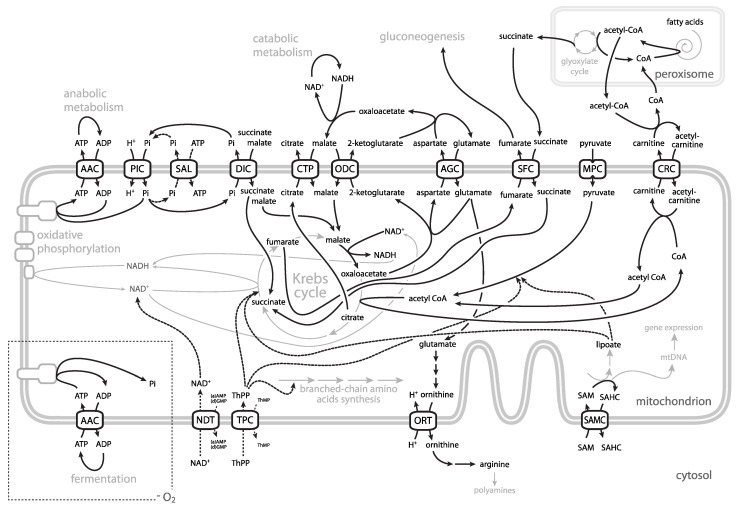
Role of mitochondrial carriers in cell metabolism. Participation of individual mitochondrial carriers in cellular metabolic pathways is indicated. Dashed arrows indicate transport that mediate the maintenance of mitochondrial pool of respective metabolite rather than metabolic flux. In the dotted rectangle, the transport of adenine nucleotides under anaerobic and fermentative conditions is shown. Outer mitochondrial membrane is not shown as it is permeable for all depicted solutes. Abbreviations: AAC—ADP/ATP carrier, AGC—aspartate-glutamate carrier, CTP—citrate carrier, CRC—carnitine carrier, DIC—dicarboxylate carrier, MPC—pyruvate carrier, NDT—NAD^+^ carrier, ODC—oxodicarboxylate carrier, ORT—ornithine carrier, PIC—phosphate carrier, SAL—ATP-Mg/P_i_ carrier, SAMC—S-adenosylmethionine (SAM) carrier, SFC—succinate/fumarate carrier, TPC—thiamine pyrophosphate carrier, SAHC—S-adenosylhomocysteine, SAM—S-adenosylmethionine, ThMP—thiamine monophosphate, ThPP—thiamine pyrophosphate.

**Figure 2 microorganisms-09-02044-f002:**
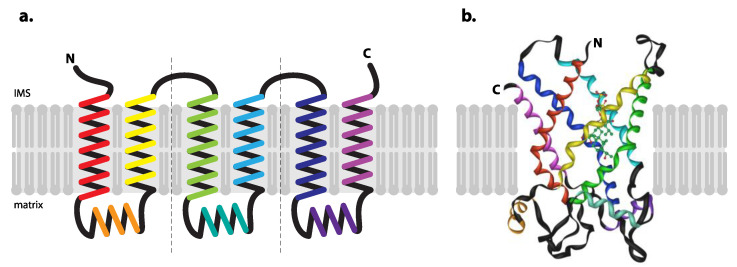
Structure of transporters of the mitochondrial carrier family. Mitochondrial carriers typically consist of three repetitive analogous modules. Each module, roughly 100 amino acids in length, contains two transmembrane α-helices in the matrix side connected with another short α-helix. Short hydrophilic loops connecting individual repetitive modules as well as both C- and N-termini of the protein are located in the intermembrane space (IMS) [47,48]. Depicted are (**a**) a schematic of membrane topology and (**b**) a 3D structure of bovine ADP/ATP carrier (AAC1). Same colour coding is used in both panels. The 3D model was generated by iMol 0.40 software using structural data from [49]. Molecule of the inhibitor, carboxyatractyloside, bound to the carrier is shown as a balls-and-sticks model.

**Table 1 microorganisms-09-02044-t001:** Overview of identified yeast mitochondrial carriers, their human homologues and associated human diseases.

Mitochondrial Carrier	Yeast Genes ^1^	Human Genes ^1^	Human Disease (OMIM) ^2^
ADP/ATP	*AAC1* *AAC2* (*PET9*) *AAC3*	ANT1 (SLC25A4) ANT2 (SLC25A5) ANT3 (SLC25A6) ANT4 (SLC25A31)	autosomal dominant progressive external ophthalmoplegia with mtDNA deletions (adPEO) (609283); mtDNA depletion syndrome 12a (cardiomyopathic type), autosomal dominant, MTDPS12a (617184); mtDNA depletion syndrome 12b (cardiomyopathic type), autosomal recessive, MTDPS12b (615418)
ATP-Mg/P_i_	*SAL1*	APC1 (SCaMC-1, SLC25A24) APC2 (SCaMC3, SLC25A23) APC3 (SCaMC2, SLC25A25) SCaMC-3-like (SLC25A41)	Fontaine progeroid syndrome (612289)
phosphate	*MIR1* *PIC2*	PiC (SLC25A3)	mitochondrial phosphate carrier deficiency, MPCD (610773)
thiamine pyrophosphate	*TPC1*	TPC (SLC25A19)	Amish microcephaly (607196); bilateral striatal necrosis (613710)
ornithine/citrulline	*ORT1* (*ARG11*)	ORC1 (Ornt1, SLC25A15) ORC2 (Ornt2, SLC25A2)	hyperornithinaemia-hyperammonaemia-homocitrullinuria (238970)
aspartate/glutamate	*AGC1*	AGC1 (aralar1, SLC25A12) AGC2 (citrin, SLC25A13)	developmental and epileptic encephalopathy 39 (612949); neonatal onset type 2 citrullinaemia (NiCCD) (605814); adult-onset type 2 citrullinaemia (CTLN2) (603471)
oxodicarboxylate	*ODC1* *ODC2*	ODC (SLC25A21)	mtDNA depletion syndrome and spinal muscular atrophy-like disease (618811)
dicarboxylate	*DIC1*	DIC (SLC25A10)	intractable epileptic encephalopathy with complex I deficiency
citrate	*CTP1*	CIC (SLC25A1)	combined d-2- and l-2-hydroxyglutaric aciduria (615182)
NAD^+^	*NDT1* *NDT2* (*YEA6*)	SLC25A51 (MCART1) SLC25A52 (MCART2)	
S-adenosylmethionine	*SAM5* (*PET8*)	SAMC (SLC25A26)	combined oxidative phosphorylation deficiency 28, COXPD28 (616794)
coenzyme A	*LEU5*	SLC25A42	recurrent metabolic crises with variable encephalomyopathic features and neurologic regression, MECREN (618416)
succinate/fumarate	*SFC1* (*ACR1*)		
citrate/oxoglutarate	*YHM2*		
oxaloacetate	*OAC1*		
carnitine	*CRC1*	CAC (SLC25A20)	carnitine-acylcarnitine translocase deficiency (212138)
glycine	*HEM25* *YMC1?*	GlyC (SLC25A38)	sideroblastic anemia-2 (SIDBA2), (pyridoxine-refractory) (205950)
glutamate	*YMC2*	GC1 (SLC25A22) GC2 (SLC25A18)	developmental and epileptic encephalopathy 3 (609304)
FAD	*FLX1*		
GTP/GDP	*GGC1* (*YHM1*, *SHM1*)		
pyrimidine nucleotides	*RIM2*	PNC1 (SLC25A33) PNC2 (SLC25A36)	
APS, PAPS ^3^	*MRX21*	SLC25A42	
Fe^2+^	*MRS3* *MRS4*	MFRN1 (SLC25A37) MFRN2 (SLC25A28)	
Mg^2+^ exporter	*MME1*		
pyridoxal 5′-phosphate	*MTM1*		
uncharacterized	*MRX20*		
ATP/AMP	*ANT1* ^4^		
	*UGO1* ^5^		
pyruvate ^6^	*MPC1* *MPC2* *MPC3*	MPC1 MPC2	mitochondrial pyruvate carrier deficiency, MPYCD (614741)

^1^ alternative designations are shown in parentheses. ^2^ number of diseases in Online Mendelian Inheritance in Man database. ^3^ adenosine-5′-phosphosulphate, 3′-phosphoadenosine-5′-phosphosulphate. ^4^ peroxisomal carrier. ^5^ despite homology, not a mitochondrial carrier. ^6^ not MCF member.

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
