# Peer review of "Learning from Yeast about Mitochondrial Carriers"

_microorganisms, 2021, doi:10.3390/microorganisms9102044_

Round 1
Reviewer 1 Report
Great to see more researchers use yeast as tools to investigate important scientific questions. According to the title, I expected to know more how to use yeast as a tool to study disease associated mitochondria carriers. This article describe relatively comprehensive contents about mitochondria carriers, not too much about how to use yeast to study it. Using yeast to study this issue mostly about the mitochondria respiration/ metabolism; therefore more comprehensive and detailed explanations about aerobic growth, anaerobic growth, Crabtree effects, diauxic shift should be provided in the context around Line 90. And the description about mitochondria carriers should be brief and focus on how to design and interpret the results of yeast experiments.
Author Response
Reviewer 1:
Great to see more researchers use yeast as tools to investigate important scientific questions. According to the title, I expected to know more how to use yeast as a tool to study disease associated mitochondria carriers. This article describe relatively comprehensive contents about mitochondria carriers, not too much about how to use yeast to study it. Using yeast to study this issue mostly about the mitochondria respiration/ metabolism; therefore more comprehensive and detailed explanations about aerobic growth, anaerobic growth, Crabtree effects, diauxic shift should be provided in the context around Line 90. And the description about mitochondria carriers should be brief and focus on how to design and interpret the results of yeast experiments.
Authors would like to thank the reviewer for reviewing the manuscript and for the comments. As suggested by the reviewer, paragraphs describing yeast energetic metabolism, including Crabtree effect and diauxic shift, have been introduced in the way, that this information can be exploited later in the text. We tried to be careful in doing the changes in this section, as the other reviewer required us to remove repetitions from this section. We believe that this helped to improve the manuscript substantially.
This reviewer also suggests that the description of carriers should be brief and focus on how to design and interpret the results of yeast experiments. We believe that the strategy how to study mitochondrial carriers in yeast is described throughout the text using individual carriers as examples. There is not one general way how to accomplish these types of experiments. For each carrier we describe what is its metabolic function in yeast and what are the phenotypes associated with its deletion (in yeast) and other relevant aspects. Human genes are described together with diseases, with which they are associated, and studies on them using yeast as a model are reviewed. Although we understand that there are other ways how to review this topic, we believe that the form that we use is effective and that manuscript in this form will be acceptable for the reviewer.
Reviewer 2 Report
The manuscript by M. Mentel et al. is a comprehensive and updated review that will attract attention of a large number of scientists. The authors have highlighted the importance of the study of mitochondrial carriers in yeast for the identification of their human orthologs and for our understanding of their associated human diseases. I recommend the publication of this review provided that the comments reported below are dealt with properly.
- section 2 “mitochondrial metabolism and genetics”: this section contains several repetitions that should be removed.
- section 3 “ADP/ATP carrier”: the authors should briefly mention that, before all the work on the molecular identification of this carrier carried out in yeast and well discussed by them, Klingenberg (Biochim. Biophys. Acta 1965, 104, 312-315) and Vignais (Biochim. Biophys. Acta 1965, 107, 184-188) in 1965 had hypothesized the existence of a membrane component of protein nature to explain the ATP/ADP exchange activity they experimentally demonstrated using rat liver mitochondria.
- lines 174-175 “3-D structure that and generalized 174 model carrier …”: reword !
- line 174: I am not convinced that the current information about mitochondrial carrier topology comes only from studies with yeast proteins. Actually it derives mainly from experiments conducted with mammalian mitochondrial carriers (see for example Biochemistry 1991, 30, 4963-4969; Biochemistry 1994, 33, 3705-3713; FEBS Lett. 1995, 357, 297-300; J. of Bioenergetics and Biomembranes 1993, 25, 493-501).
- lines 179-180: the description of the “three analogous parts” of the mitochondrial carriers is not satisfactory. It should be re-written and completed.
- lines 512-513: to avoid confusion the authors should correct the name “PiC 2” for the human phosphate carrier. It is generally known as “PiC”.
- lines 514-516: the amounts of the two isoforms of PiC and their tissue distribution have been determined in JBC 1998, 273, 22782-22787. This reference has, therefore, to be added in line 516. Moreover, in the same paper important function properties of the two PiC isoforms have been reported which are relevant for the comprehension of the PiC associated disease.
- lines 622-623 : throughout the review the authors stressed the importance of the study of yeast mitochondrial carriers for the molecular identification of their orthologs in humans. However, there is at least one example (the aspartate glutamate carrier, AGC) where identification of the carrier in humans aided identification of its ortholog in yeast. This information should be incorporated in the review.
- lines 661-662 “Two isoforms of the AGC exist in human cells, AGC1 (named aralar1, encoded by 661 SLC25A12) and AGC2 (named citrin, encoded by SLC25A13).”: this general statement derives from the experiments performed in EMBO J. 2001, 20, 5060-5069, and in fact it is exactly the conclusion of the EMBO J. 2001 paper. This reference should, therefore, be added at the end of the above mentioned sentence.
- lines 667-668 “As in yeast, human AGC, together with oxoglutarate carrier, plays a crucial role in …”: In yeast the malate aspartate shuttle requires the activity of the mitochondrial carriers AGC and ODC1-2, whereas in mammalians the activity of AGC1-2 and oxoglutarate carrier. As the latter carrier has been mentioned incidentally only in this sentence even without an appropriate reference (line 668), a confusion between yeast ODC1-2 (extensively discussed) and the mammalian oxoglutarate carrier may be generated in the readers. Note that yeast ODC1-2 and the mammalian oxoglutarate carrier in the malate aspartate shuttle function identically by catalyzing an oxoglutarate/malate exchange, and (to make the confusion even more likely) an ortholog of the yeast ODC1-2 (able to transport oxoglutarate but not malate) exists in mammalians and is also named ODC. To clarify this issue, the authors should at least add references immediately after the words “together with oxoglutarate carrier” (lines 667-668). Appropriate references are Biochim. Biophys. Acta 1985, 810, 362-369; Biochemistry 1990, 29, 11033-11040; Biochem. J. 1993, 294, 293-299; Biochim. Biophys. Acta 2011, 1807, 302-310.
- lines 676-678 “Mutations in the SLC25A12 gene causing defects in the brain-dominant isoform 676 AGC1 lead to rare neurological disease (OMIM 612949) connected with global cerebral 677 hypomyelination, developmental delay, epilepsy and hypotonia.”: appropriate references are missing at the end of this important sentence. The first mutations in the SLC25A12 gene have been described in NEJM 2009, 361, 489-495 and JIMD Reports 2014,14, 77-85; at least these papers should, therefore, be cited. Rather strangely, the authors have added a ref. to the subsequent less important sentence (lines 678-679): “Moreover, it seems that 678 there is a certain link between impaired AGC1 function and autism [126].”
- lines 682-687: the paper (precedent to that cited as ref. 127) showing that a ketogenic diet is beneficial to AGC1 deficiency should be cited (M. Dahlin et al. Epilepsia 2015, 56, e176-e181).
- lines 772-773 : in this respect the authors should consider the papers published (in Oncotarget 2012, 3, 1220-… and Oncotarget 2014, 5, 1212-1225) before than that cited as ref. 141.
- lines 906-912 are orphan of the appropriate reference which is the already cited ref. 161.
- lines 948-950 : the sentence should be reworded.
- lines 997-999 : the sentence should be reworded.
- line 1058 : to avoid confusion in the readers’ mind the acronysm CAC (often used in the field) should be added.
- line 1065 : perhaps a “the” is missing before the words “mitochondrial carrier”.
- line 1080 : it should be “14. Other yeast mitochondrial carriers” and not “14. Other mitochondrial carriers”.
- lines 1082-1085 “ … carriers specific for the transport of Coenzyme A, succinate/fumarate, citrate/oxoglutarate, oxalacetate, glycine, FAD, …” : these are other important mitochondrial carriers in yeast which have not been discussed and have only been mentioned in Table I. The authors should provide the references of the works in which they have been first characterized (as the authors have done for the mitochondrial carriers discussed in the review).
Author Response
Reviewer 2:
The manuscript by M. Mentel et al. is a comprehensive and updated review that will attract attention of a large number of scientists. The authors have highlighted the importance of the study of mitochondrial carriers in yeast for the identification of their human orthologs and for our understanding of their associated human diseases. I recommend the publication of this review provided that the comments reported below are dealt with properly.
We would like to thank the reviewer for thorough review. We recognize that addressing the reviewer concerns improved the manuscript substantially. The changes in the manuscript are listed below as a point-by-point answer.
- section 2 “mitochondrial metabolism and genetics”: this section contains several repetitions that should be removed.
We did changes to this section. As the other reviewer suggested to describe several aspects of yeast metabolism, we have done so and made small changes in this section. We believe that if there are some repetitions left, these are the minimum and help keep text easier to follow.
- section 3 “ADP/ATP carrier”: the authors should briefly mention that, before all the work on the molecular identification of this carrier carried out in yeast and well discussed by them, Klingenberg (Biochim. Biophys. Acta 1965, 104, 312-315) and Vignais (Biochim. Biophys. Acta 1965, 107, 184-188) in 1965 had hypothesized the existence of a membrane component of protein nature to explain the ATP/ADP exchange activity they experimentally demonstrated using rat liver mitochondria.
We have changed the sentence in which we describe how the first yeast gene encoding ADP/ATP carrier was identified. The sentence now reads “Analyses of mitochondria isolated from the op1 mutant cells, including the response of parameters of respiration to bongkrekic acid and atractyloside, specific inhibitors of mitochondrial ADP/ATP transport, indicated that the mutation may affect the ADP/ATP translocator of inner mitochondrial membrane [23, 24], a protein whose existence in mitochondria was suggested earlier, when transport of ATP and ADP was initially investigated using rat liver mitochondria [25, 26].”. References 25 and 26 are the two mentioned by the reviewer.
- lines 174-175 “3-D structure that and generalized 174 model carrier …”: reword !
We apologize for the mistake introduced by erroneous editing. Mistake was removed as the sentence was rephrased in order to address the concern 4.
- line 174: I am not convinced that the current information about mitochondrial carrier topology comes only from studies with yeast proteins. Actually it derives mainly from experiments conducted with mammalian mitochondrial carriers (see for example Biochemistry 1991, 30, 4963-4969; Biochemistry 1994, 33, 3705-3713; FEBS Lett. 1995, 357, 297-300; J. of Bioenergetics and Biomembranes 1993, 25, 493-501).
We rephrased the statement to reflect the concern of reviewer, which is indeed correct. The sentence now reads “While the first hints on membrane topology of mitochondrial carriers came from analyses of proteins isolated from mammalian mitochondria [40-44], functional analyses of yeast AACs and ultimately the crystallization of bovine protein led to the detail model of its membrane topology, 3-D structure that and generalized model carrier of MCF (see [45, 46] for review and Figure 2).” References 40-44 are the ones mentioned by the reviewer.
- lines 179-180: the description of the “three analogous parts” of the mitochondrial carriers is not satisfactory. It should be re-written and completed.
The sentence was replaced by following text “Mitochondrial carriers typically consist of three repetitive analogous modules. Each module, roughly 100 amino acids in length, contains two transmembrane α-helices in the matrix side connected with another short α-helix. Short hydrophilic loops connecting individual repetitive modules as well as both C- and N- termini of the protein are located in the intermembrane space (IMS) [47, 48].”
- lines 512-513: to avoid confusion the authors should correct the name “PiC 2” for the human phosphate carrier. It is generally known as “PiC”.
Corrected as suggested by the reviewer.
- lines 514-516: the amounts of the two isoforms of PiC and their tissue distribution have been determined in JBC 1998, 273, 22782-22787. This reference has, therefore, to be added in line 516. Moreover, in the same paper important function properties of the two PiC isoforms have been reported which are relevant for the comprehension of the PiC associated disease.
The reference has been added together with text that explains what we believe the reviewer meant. The text now reads “The same tissue distribution has been reported for bovine isoforms [116]. The different enzymatic properties of the isoforms suggest that while the role of isoform B is to provide mitochondria in all tissues with basal phosphate uptake, the isoform A provides additional transport in the high energy demanding tissues [116].” The reference 116 is the one mentioned by the reviewer.
- lines 622-623 : throughout the review the authors stressed the importance of the study of yeast mitochondrial carriers for the molecular identification of their orthologs in humans. However, there is at least one example (the aspartate glutamate carrier, AGC) where identification of the carrier in humans aided identification of its ortholog in yeast. This information should be incorporated in the review.
We incorporated the sentence that emphasizes the mentioned fact. The sentence reads “Interestingly, in this case the identification of yeast carrier followed the characterization of human orthologues [130].”.
- lines 661-662 “Two isoforms of the AGC exist in human cells, AGC1 (named aralar1, encoded by 661 SLC25A12) and AGC2 (named citrin, encoded by SLC25A13).”: this general statement derives from the experiments performed in EMBO J. 2001, 20, 5060-5069, and in fact it is exactly the conclusion of the EMBO J. 2001 paper. This reference should, therefore, be added at the end of the above mentioned sentence.
The reference has been added. It is reference 136.
- lines 667-668 “As in yeast, human AGC, together with oxoglutarate carrier, plays a crucial role in …”: In yeast the malate aspartate shuttle requires the activity of the mitochondrial carriers AGC and ODC1-2, whereas in mammalians the activity of AGC1-2 and oxoglutarate carrier. As the latter carrier has been mentioned incidentally only in this sentence even without an appropriate reference (line 668), a confusion between yeast ODC1-2 (extensively discussed) and the mammalian oxoglutarate carrier may be generated in the readers. Note that yeast ODC1-2 and the mammalian oxoglutarate carrier in the malate aspartate shuttle function identically by catalyzing an oxoglutarate/malate exchange, and (to make the confusion even more likely) an ortholog of the yeast ODC1-2 (able to transport oxoglutarate but not malate) exists in mammalians and is also named ODC. To clarify this issue, the authors should at least add references immediately after the words “together with oxoglutarate carrier” (lines 667-668). Appropriate references are Biochim. Biophys. Acta 1985, 810, 362-369; Biochemistry 1990, 29, 11033-11040; Biochem. J. 1993, 294, 293-299; Biochim. Biophys. Acta 2011, 1807, 302-310.
The references have been added as suggested by the reviewer. They are references 138-141.
- lines 676-678 “Mutations in the SLC25A12 gene causing defects in the brain-dominant isoform 676 AGC1 lead to rare neurological disease (OMIM 612949) connected with global cerebral 677 hypomyelination, developmental delay, epilepsy and hypotonia.”: appropriate references are missing at the end of this important sentence. The first mutations in the SLC25A12 gene have been described in NEJM 2009, 361, 489-495 and JIMD Reports 2014,14, 77-85; at least these papers should, therefore, be cited. Rather strangely, the authors have added a ref. to the subsequent less important sentence (lines 678-679): “Moreover, it seems that 678 there is a certain link between impaired AGC1 function and autism [126].”
The references have been added. They are references 145 and 146.
- lines 682-687: the paper (precedent to that cited as ref. 127) showing that a ketogenic diet is beneficial to AGC1 deficiency should be cited (M. Dahlin et al. Epilepsia 2015, 56, e176-e181).
The reference has been added (148).
- lines 772-773 : in this respect the authors should consider the papers published (in Oncotarget 2012, 3, 1220-… and Oncotarget 2014, 5, 1212-1225) before than that cited as ref. 141.
The reference have been added (163 and 164).
- lines 906-912 are orphan of the appropriate reference which is the already cited ref. 161.
The reference has been added.
15. lines 948-950 : the sentence should be reworded.
The sentence was reworded. It now reads: “When identified in S. cerevisiae it was originally assigned a designation Arg11p to reflect its role in the biosynthesis of arginine [191], later a designation Ort1p became widely accepted.”
- lines 997-999 : the sentence should be reworded.
The sentence was reworded. It now reads: “Of the tested single substitutions, the growth of the ∆ort1 strain was only rescued by the Orc2p Gln179Arg. The growth was further improved by combined substitutions Gln179Arg/Met72Ile or Gln179Arg/Ala123Val.”
- line 1058 : to avoid confusion in the readers’ mind the acronysm CAC (often used in the field) should be added.
The acronym was introduced as suggested.
- line 1065 : perhaps a “the” is missing before the words “mitochondrial carrier”.
Article “the” was inserted as suggested.
- line 1080 : it should be “14. Other yeast mitochondrial carriers” and not “14. Other mitochondrial carriers”.
Title was changed as suggested by the reviewer.
- lines 1082-1085 “ … carriers specific for the transport of Coenzyme A, succinate/fumarate, citrate/oxoglutarate, oxalacetate, glycine, FAD, …” : these are other important mitochondrial carriers in yeast which have not been discussed and have only been mentioned in Table I. The authors should provide the references of the works in which they have been first characterized (as the authors have done for the mitochondrial carriers discussed in the review).
References have been added to each mentioned mitochondrial carrier. They are references 211-221.
Round 2
Reviewer 2 Report
The revised manuscript “Learning from yeast about mitochondrial carriers” by M. Mentel et al. has been considerably improved. However, before this manuscript can be accepted for publication the following minor comments (which are related to remarks previously made by this reviewer) should be dealt with properly.
- lines 658-659: insert the words “identification and” between the words “… yeast carrier followed the” and “characterisation of human orthologues”
- lines 940-942: the authors have overlooked the previously made remark that “these lines are orphan of the appropriate reference” which is (in the revised manuscript) ref. 185.
Author Response
Dear reviewer,
Thank you very much for thorough review. We apologise for these mistakes.
In this revision we have done as you suggested. We inserted words "identification and" in line 659 and inserted reference 185 into line 942.
Yours sincerely
Peter Polcic